



# Linking catchment hydrology and ocean circulation in Late Holocene southernmost Africa

Annette Hahn[1]; Enno Schefuß[1]; Sergio Andò[2], Hayley C. Cawthra[3,4], Peter Frenzel[5];Martin Kugel[1], Stephanie Meschner[5]; Gesine Mollenhauer[6]; Matthias Zabel[1]

[1] MARUM Center for Marine Environmental Sciences, University of Bremen, Bremen, Germany
[2] Department of Earth and Environmental Sciences, University of Milano-Bicocca, Milan, Italy
[3] Geophysics Competency, Council for Geoscience, Cape Town, South Africa
[4] Centre for Coastal Palaeoscience, Nelson Mandela Metropolitan University, Port Elizabeth, South Africa
[5] Institute of Earth Sciences; Friedrich Schiller University of Jena, Germany
[6] Alfred-Wegener-Institute, Bremerhaven, Germany

*Correspondence to*: Annette Hahn (ahahn@marum.de)

**Abstract.** Recent studies of the South African climatic system revealed a highly complex interplay of forcing factors on precipitation regimes. This includes the influence of the tropical easterlies, the strength of the Southern Hemispheric

Westerlies as well as sea surface temperatures along the coast of the subcontinent. This study of a sediment core at the terrestrial-marine interface spanning a time interval of ~4 ka provides insights on the highly dynamic climatic system in southernmost South Africa. Several organic proxies sensitive to changes in climatic parameters like the distribution and isotopic composition of plant-wax lipids as well as indicators for sea surface temperatures and soil input give information on climatic changes during the investigated time period. Moreover, the micropaleontology, mineralogical and elemental

composition of the sediments reflects the variability of the terrigenous input to the core site. The combination of downcore sediment signatures and a catchment-wide provenance study indicate that the Little Ice Age was characterized by relatively warm sea surface temperatures in Mossel Bay and arid climatic conditions favorable to torrential flood events sourced in the Gouritz headlands. In contrast, the so-called Medieval Climate Anomaly is expressed by humid conditions in the Gouritz River catchment with lower, but highly variable sea surface temperatures in the Mossel Bay area. The coincidence between

humid climatic conditions and cooler sea surface temperatures has been attributed to a strengthened and more southerly anticyclonic circulation. In this climatic setting strong tropical easterlies supply Indian Ocean moisture to South Africa and at the same time Agulhas Bank upwelling pulses become more common due to an increase in Agulhas Current transport as well as alongshore southeasterly winds. These processes resemble the modern day oceanography in summer and can be conceptualized in a regional climate model.



## 1 Introduction

South Africa's geographic position plays an important role in unravelling global climate history, in that Atlantic Ocean Circulation can be constrained via the Atlantic meridional overturning circulation (Toggweiler and Lea, 2010; Beal et al., 2011). Moreover, South Africa's regional climate is expected to be particularly sensitive to future climate change (Metz,

2007; Kirtman et al., 2013).South Africa's precipitation patterns are influenced by shifts of the Southern Hemispheric Westerlies (SHW), tropical easterly strength, as well as variations in ocean circulation patterns (e.g. Chase and Meadows, 2007; Chase et al., 2011; Marzin and Braconnot, 2009; Schefuß et al., 2011; Fig. 1). The interplay and regional extent of these factors and their relation to global climate forcings at different time scales is however, far from being understood. While wet phases in western South Africa's winter rainfall zone (WRZ) have been linked to northward shifts of the Antarctic

circumpolar circulation and the resulting strengthening of the SHW in several studies (Stuut et al., 2004; Chiang and Bitz, 2005; Stager et al., 2012; Chase et al., 2013), van Zinderen-Bakker (1976; 1978; 1983) proposed a model suggesting that the South Coast would receive less winter rainfall during glacials. From isotopic analysis of Late Pleistocene speleothem at Mossel Bay, Bar-Matthews et al. (2010) concurred that when proxies suggest warmer global temperatures, more winter rainfall and C3 grasses were expected on the South Coast. An overall expansion of winter rainfall in response to global

cooling was not supported by these results. A further dispute currently concerns the forcings behind the easterly rainfall regimes in the southernmost part of Africa, which is not influenced by modern ICTZ shifts (south of ~13-14°S). The major question is whether the climate variability in southern Africa is synchronous with northern hemispheric variations or driven by direct insolation changes and thus anti-phased to northern hemispheric signals? It was relatively uncontroversial that long term (glacial-interglacial timescale) climate variations in southernmost Africa are directly forced by local insolation

(Partridge et al., 1997) until recent paleorecords have emerged in which the direct insolation is not evident as a climatic driver (Chase et al., 2009, 2010; Dupont et al., 2011) or opposing trends (i.e. synchrony with the northern hemisphere) were observed (Tierney et al., 2008; Stager et al., 2011; Truc et al., 2013). Furthermore, on shorter timescales, recent datasets have revealed additional complexity within the system: Eastern South African studies from Wonderkrater (Truc et al., 2013) and regional pollen analyses (Scott et al., 2012), show a highly complex system of enhanced and diminished precipitation with

respect to the geographical position of the archive. A progressive Holocene aridification in northwestern Namibia (Austerlitz midden, Chase et al., 2010), consistent with a proposed cool and dry period in eastern South Africa (deduced from the Cold Air Cave stalagmite record) (Holmgren et al., 1999) as well as Braamhoek peatland record from the Drakensberg Mountains (Norström et al., 2014) and a regional compilation of pollen records (Chevailier and Chase, 2015) all offer additional evidence for climate variability in phase with the northern hemisphere. Mechanisms suggested to explain the transmission of

the northern hemispheric signal to southern Africa include tele-connections causing a dipole-pattern of rainfall between eastern tropical and southern Africa (Norström et al., 2014) as well as sea surface temperature (SST) changes due to ocean circulation variability (Agulhas strength) (Tierney et al., 2008; Stager et al., 2011; Scott et al., 2012; Truc et al., 2013). Ocean circulation patterns have been suggested to influence precipitation in South Africa on both the east and the west



coasts. In western South Africa paleoenvironmental reconstructions suggest that strong SHW reduce the leakage of warm, saline Agulhas water into the southern Atlantic, weakening the Benguela upwelling system (Kim et al., 2003; MacKellar et al., 2007; Biastoch et al., 2009a). Modelled data show similar outputs predicting a weakening of the Agulhas Current and the leakage of warm water due to northward displacement of the SHW (MacKellar et al., 2007; Biastoch et al., 2009b; Durgadoo

et al., 2013).The coevally decreased SSTs on the south western coast of South Africa may have led to decreasing precipitation in the YRZ (Rouault et al., 2003). In turn, positive SST anomalies along the east coast associated with Agulhas strengthening were suggested to enhance summer precipitation in the east South African SRZ (Jury et al., 1993; Dupont et al., 2011; Scott et al., 2012). However, further south along the east coast, the oxygen isotope composition of marine mollusk shells preserved in Nelson Bay archaeological cave deposit indicate that during periods of wetter conditions over the

southern African interior, the Agulhas surface water tempertaures were actually lower than during arid periods (Cohen and Tyson, 1995). Based on this data in combination with interannual observations a conceptual model relating oceanic and atmospheric circulation systems of southern Africa was proposed in 1995 by Cohen and Tyson. This model predicts that during periods of stronger anticyclonic circulation, the increased alongshore winds caused coastal upwelling in Mossel Bay whereas the increased Agulhas strength drove upwelling over the east coast shelf edge. The database behind this model is

however sparse and a large gap exists for the climatically interesting period between 2400 and 650 BP (Tyson and Lindesay, 1992; Cohen and Tyson, 1995). Our sediment core GeoB18308-1, located on the South Coast of South Africa, in the catchment of the Gouritz River, 30 km west of the town Mossel Bay, aims to close this gap. In addition to providing a SST reconstruction for the past ~4 ka, this record in the Gouritz River mouth also holds the potential for high resolution continental climatic reconstructions. As the Gouritz River catchment is in a unique geographical position at the interface of

westerly and easterly climate regimes, it is particularly susceptible to shifts in circulation systems occurring on shorter as well as on geologic timescales. In order to decipher the complexity of the terrigenous climatic signal, sediment provenance and transport processes are studied using catchment material in a source to sink approach.

## 2 Regional setting

### 2.1 Gouritz River Catchment Area

The Gouritz River catchment area is divided into 4 sub-catchments.The first includes the Buffels, Touws and Groot Rivers, the second consists of the Gamka and Dwyka Rivers, the third comprises the Traka, Olifants and Kammanassie Rivers, and the fourth represents the Gouritz River itself (Le Maitre et al., 2009). The mean annual runoff of the sub-catchments was measured to be $105*10^6$ m$^3$, $206*10^6$m$^3$, $229*10^6$m$^3$ and 134 Mm$^3$, respectively. The entire Gouritz River catchment is located within the year-round rainfall zone (YRZ) (Fig. 1). Le Maitre et al. (2009), however, note that even though observed

rainfall patterns in the catchment area showed year round precipitation, most of the area mainly received rainfall during summer and autumn. Measurements by Desmet and Cowling (1999) revealed highly variable mean annual precipitation, but also showed somewhat lower rainfall quantities in the winter. Sealy (1986) indicated mixed summer and winter rainfall




during the Last Glacial Maximum at Nelson Bay Cave, 120 km east of the Gouritz River. Several major floods, e.g. in January 1981, March 2003 and in March 2004, caused by extremely high rainfall events, are another characteristic of this area (Desmet and Cowling, 1999; Cowling et al., 2004). Despite the relatively small size of the catchment area (approximately 45,715 km$^2$), the altitudinal gradient is steep as the Swartberg Mountains rise abruptly above 2000m a.s.l.

within 100km of the coast (Le Maitre et al, 2009).

## 2.2 Vegetation in the study area

Within the study area three dominant vegetation types are described: Fynbos, Succulent Karoo and the Nama Karoo biomes (Mucina and Rutherford, 2006; see map in Fig. 2). The Fynbos biome is characterized as a Mediterranean-type vegetation with extraordinary diversity and endemism (Goldblatt and Manning, 2002). It is found especially along the southern and

southwestern coast of South Africa and therefore, receives most of its rainfall during austral winter. The most dominant photosynthetic pathway reported is $C_3$ with some species using the crassulacean acid metabolism (CAM) (Vogel et al., 1978). The Succulent Karoo biome is described as being better adapted to arid conditions and higher summer temperatures than the Fynbos vegetation type (Carr et al., 2014). Due to the geographic distribution of vegetation, the Succulent Karoo biome gets most of its rainfall during winter but also receives summer precipitation in the eastern part of the catchment. Here

the dominant photosynthetic pathway is described as CAM (Rundel et al., 1999). The last vegetation type, Nama Karoo, is described to consist mostly of $C_4$ grasses and due to its occurrence in a north-eastern geographical position, receives dominantly summer rain (Le Maitre et al., 2009). Studies within this region as well as in other areas of Africa revealed that these vegetation types showed differences in their *n*-alkane composition (Rommerskirchen et al., 2006; Vogts et al., 2009; Carr, 2012; 2014; Boom et al., 2014; Fig. 2).

## 2.3 Geology of the Gouritz River catchment

The distinctive topography of Southern Africa is characterized by a high-relief inland plateau flanked by a low average elevation coastal plain and the Gouritz River catchment drains almost all sequences of the Cape. The geological history of this region spans the last 560 Myr from the Neoproterozoic sequences to recent unconsolidated beach and dune sediments flanking the modern shoreline. The oldest 'basement' rocks comprise the Malmesbury Group and the Cape Granite Suite

(~550 – 510 Ma), is related to Pan African orogenesis during the formation of Gondwana (Rozendaal et al., 1999; Johnson et al., 2006; Milani and de Wit, 2008). Overlying deposits of the Cape Supergroup form a 6 – 10 km thick siliciclastic sequence, divided into the Table Mountain, Bokkeveld and Witteberg Groups (Thamm and Johnson, 2006). The lower Table Mountain Group sandstones were deposited along a regionally subsiding shelf from ~500 Ma and are overlain by middle Table Mountain Group glaciogenic deposits (Rust and Theron, 1964). Increased downwarping facilitated the deposition of

the Devonian Bokkeveld Group shales during a marine transgression and the overlying Witteberg Group sandstones and mudstones represent the last phase of shelf sedimentation prior to the onset of the ~300 – 350 Ma Dwyka glaciation and  the deformation leading to the Cape Orogeny (~278 – 230 Ma; Newton et al., 2006) producing the Cape Fold Belt (CFB).





Deposits of the Karoo Supergroup were laid down in a foreland basin adjacent to this orogeny and Karoo sedimentation was terminated by the extrusion and intrusion of the extensive Drakensberg basalts and dolerites, respectively (~183 Ma, Duncan et al., 1997). The fragmentation of Gondwana and the opening of the South Atlantic commenced in the Early Cretaceous (~136 Ma) as South America was rifted westward along the Falkland Agulhas Fracture Zone (Martin and Hartnady, 1986; Eagles, 2007). Compressional CFB faults were subsequently reactivated as listric extensional faults (Paton et al., 2006), and the newly created accommodation space in basins was filled with clastic sediments. Offshore, arcuate normal faults bounded several graben and half graben structures that became the depocentres for terrigenous sediment sourced from the onshore areas (Tinker et al., 2008a,b). The southern cape continental shelf and low relief coastal plain comprise the submerged and emergent portions of a continuous feature, the degree of separation being dependent upon the relative sea level at any given time (Cawthra et al., 2015). This broad, shallow, plain is mantled with Pleistocene/Holocene deposits. The south coast is characterized by a series of eastward-opening log-spiral bays that extend for approximately 20-40 km between adjacent west-east trending rocky headlands. The Gouritz River is associated with a well-developed, stratified, sediment wedge (~10 km wide and 85 km long) (Birch, 1978) that extends predominantly westwards of the river mouth. Offshore, the Gouritz River is associated with a subdued incised valley, running across the continental shelf (Cawthra, 2014).

## 2.4 Oceanic circulation on the eastern Agulhas Bank

The oceanography of the Agulhas Bank is strongly influenced by the warm, fast-flowing Agulhas Current originating in the Mozambique Channel (Lutjeharms et al., 2001). The inner shelf targeted in this study is influenced by wind-driven coastal upwelling, particularly during summer (c.f. Hutchings et al., 1995). Along-coast easterly winds drive these periodic events of short-term surface water cooling of up to 8°C (Schumann et al. 1982, Beckley 1983). During winter plumes of warm Agulhas current water have been observed to advect onto the shelf by southwesterly winds (Lutjeharms and van Ballegooyen, 1988). Using multibeam bathymetry and side-scan sonar Cawthra et al. (2015) describe the morphology of the wide continental shelf as low gradient and generally smooth in the environs of Mossel Bay and Vlees Bay (Fig. 3). The hard rock promontory of Cape St Blaize extends eastwards for 2.5 km and a thin veneer of Holocene sediment which blankets the inner- and mid shelf is punctuated by occasional Pleistocene deposits, which reach a maximum height of 8 m.

## 3 Material and Methods

### 3.1 Sediment Coring and catchment samples

Sediment core GeoB18308-1 was taken with a vibrocorer during RV METEOR Cruise M102 in December 2013 from a protected valley fill (Cawthra et al, 2015). The core has a total length of 4.94 m and the sampling site is located at 34°22.39'S 21°55.75'E near Mossel Bay. This shallow marine deposit (39.8 m water depth) mainly consisted of fine sand and mud, which probably was supplied to the ocean by the nearby (4 km) Gouritz River mouth. In the interval between ~30-60 cm an event deposit revealing slumped turbidite facies with an erosive contact was identified. South African sea level





reconstructions (Compton, 2006; Ramsay and Cooper, 2002) reveal that the sampling site (located in 39.8m water depth) was not significantly influenced by fluctuations during the last 5 ka. However, offshore the YRZ, the prevailing regional ocean circulation system (Agulhas Current) favors the erosion rather than the accumulation of sediment (Martin and Flemming, 1986). Holocene sediments in sufficient thickness are therefore restricted to local valley fills, within

embayments, and small-scale sediment bodies (Cawthra, 2014; Cawthra et al., 2015). Riverbank, flood deposit and suspension load (obtained by filtering ca. 100 l of water pumped from the rivers' main flow) samples were collected from 8 locations along the Gouritz River in March 2015 in order to determine the provenance of the deposited material (see locations in Fig. 2).

### 3.2 Age Model

The age model used in this study is based on 14 radiocarbon ages (Fig. 4). Ages were estimated from 9 total organic carbon (TOC) samples, 2 shells, 2 pieces of wood and 1 crab claw (Table 1). The cleaning procedures as well as the Accelerator Mass Spectrometry (AMS) measurements were carried out in the Poznań Radiocarbon Laboratory, Poland and Beta Analytic Radiocarbon Dating Laboratory Florida, USA. Depending on sediment type (terrestrial or marine identified via BIT-Index) either the Southern Hemisphere calibration curve (SHCal13) (Hogg et al., 2013) or the modelled ocean average curve

(Marine13) (Reimer et al., 2013) were used to calibrate the radiocarbon ages. The marine ΔR is assumed to be close to the south west coast ΔR (146 ± 85 14C) published in 2012 by Dewar et al. (Meadows et al. in prep./personal communication). The software Bacon (Blaauw and Christen, 2011) was used to calculate an age model from which sedimentation rates were computed. We refer to median age estimations in this paper; please note the associated uncertainty indicated in Fig. 4 and Table 1.

### 3.3 Inorganic Geochemistry

Geochemical analysis (organic and inorganic) were performed downcore (2 to 10 cm resolution) as well as on soil, riverbank, flood deposit and suspension load samples collected from 8 Gouritz catchment locations (ECT-1 to ECT-8) in March 2015 (Fig. 2). Soil, riverbank and flood deposit were collected in plastic sample bags whereas suspension load samples were obtained by filtering ~100 l of water pumped from the rivers main flow. Scanning-data was collected using

MARUM XRF Core Scanner II (AVAATECH Serial No. 2) equipped with an Oxford Instruments 50W XTF5011 rhodium X-Ray tube, a Canberra X-PIPS Silicon Drift Detector (SDD; Model SXD 15C-150-500) run at a 150eV resolution and a Canberra Digital Spectrum Analyzer DAS 1000 at the MARUM-University of Bremen. The depth resolution was 1 cm (1.2 cm$^2$, 1mm slit size) with two generator settings (30kV, 1mA, 20s; 10kV, 0.2kV, 20s) for detection of different elemental groups. For calibration purposes the XRF-spectrometer measurements were completed on 28 selected sediment samples

using a PANalytical Epsilon3-XL XRF spectrometer equipped with a rhodium tube, several filters and a SSD5 detector. Elemental counts were quantified using a calibration based on certified standard materials (GBW07309, GBW07316, MAG-1). Scanning intensities were normalized using the discrete ED-XRF measurements and the procedure as proposed by Lyle et




al. (2012). Good correlations ($R^2$ above 0.7) of XRF scanner and XRF analyses of discrete samples results were found for a variety of elements including Ca and Fe.

### 3.4 Organic Geochemistry

The continuous high resolution, XRF scanning dataset (Fig. S1) indicates little variability in the lower half of core
GeoB18308-1. Based on this finding 38 sub-samples for biomarker analysis were selected.

#### 3.4.1 Plant-wax analyses

Dried and ground samples were extracted using a DIONEX Accelerated Solvent Extractor (ASE 200) at 100°C, 1000 psi with a 9:1 mixture of dichloromethane to methanol for 5 min and 3 extraction cycles. Prior to extraction, squalane and $C_{46}$-GDGT (glycerol dialkyl glycerol tetraether) were added as internal standards. Asphaltenes were removed from the total lipid
extracts using $Na_2SO_4$ columns and elution with hexane. Extracts were saponified for 2 hours at 85°C in 500 µL of a 0.1M KOH-solution in methanol (MeOH). Neutral lipids were recovered by liquid-liquid extraction using hexane. Lipid fractions were separated over a silica gel column (10% deactivated) using hexane (hydrocarbons), DCM (ketones), and DCM:MeOH1:1 (polar fractions). Unsaturated compounds were removed from the hydrocarbon fractions by column chromatography over $AgNO_3$-coated silica using hexane. The saturated hydrocarbon fractions containing *n*-alkanes were
injected in splitless mode at 260°C into a Thermo Scientific Focus gas chromatograph equipped with a DB-5ms column (30m x 0.25mm, 0.25µm film thickness, Agilent Technologies, Palo Alto, USA) coupled to a flame ionisation detector (GC-FID). The oven was held at 70°C for 2 min, then heated at a rate of 20°C min$^{-1}$ to 150°C, and after with a rate of 4°C min$^{-1}$ to 320°C, and remained at this temperature for 16.5 min. An external calibration standard containing *n*-alkanes of known concentrations was analysed every six samples. Based on the repeated standard analyses the precision of quantification is
calculated to 5%. The ratio between $C_{29}$ and $C_{31}$ *n*-alkanes (Norm31; Schefuß et al. 2013) was calculated using Eq. (1):

$$\text{Norm31} = C_{31}/(C_{29} + C_{31}) \tag{1}$$

The carbon preference index (CPI) (Bray and Evans, 1961) was calculated according using Eq. (2):

$$\text{CPI}_{23-33} = \frac{1}{2} \left\{ \left( \frac{C_{23} + C_{25} + C_{27} + C_{29} + C_{31} + C_{33}}{C_{22} + C_{24} + C_{26} + C_{28} + C_{30} + C_{32}} \right) + \left( \frac{C_{23} + C_{25} + C_{27} + C_{29} + C_{31} + C_{33}}{C_{24} + C_{26} + C_{28} + C_{30} + C_{32} + C_{34}} \right) \right\} \tag{2}$$

#### 3.4.2 Compound Specific Carbon and Hydrogen isotope analyses

The fractions containing *n*-alkanes were used for compound-specific carbon and hydrogen isotope analyses. Compound-
specific stable hydrogen isotope measurements were performed on a Trace GC (Thermo FisherScientific, Bremen, Germany)





coupled to a MAT 253 IRMS (Finnigan MAT, Bremen, Germany) via a pyrolysis reactor operated at 1420°C. The PTV injector was maintained at 45°C at injection and then heated to 340°C to transfer the sample onto the GC column. Compounds were separated on a Rxi-5ms silica column (30m x 0.25mm, 0.25µm film thickness, Restek, Bellefonte, USA). The GC was maintained at 120°C for 3 min then heated to 200°C with 30°C min$^{-1}$, then at 4°C min$^{-1}$ to 320°C and held for

24 min. H$_2$ reference gas was used for isotope calibration. The δD values are expressed in ‰ relative to VSMOW. The H3+ factor varied around 6.12 ± 0.02 ppm nA$^{-1}$. Long-term repeated analysis of the external standard mixture with 16 *n*-alkanes rendered a precision (1σ) of ± 3 ‰ and an average accuracy of 0 ‰. The internal standard squalene had an accuracy and precision of 0 and 2 ‰, respectively.

Carbon isotope compositions of the *n*-alkanes were analysed on the same type of GC coupled to a MAT 252 IRMS via a

modified GC/C III combustion interface operated at 1000°C. Injector and GC setting were similar as for    D analysis. Calibrated CO$_2$ reference gas was used for isotope calibration. Values are expressed in $δ^{13}C$ relative to VPDB. Long-term repeated analysis of the external standard mixture with 16 *n*-alkanes rendered a precision (1σ) of ± 0.3 ‰ and an average accuracy of 0.4 ‰. The internal standard squalene had an accuracy and precision of 0.4 and 0.2 ‰, respectively.

### 3.4.3 Analysis of GDGTs

The concentrations of Glycerol-Dibiphytanyl-Glycerol-Tetraethers (GDGT) were determined at the Department of Geoscience – University Bremen. Polar fractions containing GDGTs were filtered through a 0.45 µm PTFE filter and weighted before analysis. The instrument used to determine GDGT concentrations was an Agilent 1200 series high performance liquid chromatograph (HPLC) coupled with an atmospheric pressure chemical ionization interface (APCI) to an Agilent 6120 quadrupole mass spectrometer (MS). The method was slightly modified from prior work (e.g. Hopmans et al.,

2000). GDGTs were eluted over a Prevail Cyano column (Grace, 3 µm, 150 mm×2.1 mm) maintained at 30 °C using a gradient of solvent A (n-haxane) and B (5% iosopropanol in n-hexane) from 80% solvent A (5 min) followed by a linear increase to 36% solvent B in 40 min. The flow rate was 0.2 mL min$^{-1}$. GDGTs were detected in positive-ion mode of the APCI-MS and selective ion monitoring (SIM). The APCI spray-chamber specifications were: nebulizer pressure 50 psi, vaporizer temperature 350°C, N$_2$ drying gas flow 5 l min-1 and 350 °C, capillary voltage −4kV and corona current +5 µA.

Peak areas of the target compounds were used to compute the following proxies: The TetraEther indeX TEX$_{86}^{H}$ was used as a proxy to estimate SSTs (Kim et al. 2010). The TEX$_{86}^{H}$ is based on the ratio between isoprenoidal GDGTs of 86 carbon atoms, containing 1, 2 and 3 cyclopentane moieties (I, II and III) as well as the regioisomer of crenarchaeol, a GDGT with 4 cyclopentane moiety (V´). These compounds represent membrane constituents of planktonic archaea, which were found to shift in abundance with changing SSTs (Schouten et al., 2002). TEX$_{86}^{H}$ was calculated with Eq. (3) whereas for SST

reconstruction Eq. (4) was used (Kim et al., 2010).

$$TEX_{86^{H}} = \log \frac{[II+III+V´]}{[I+II+III+V´]} \tag{3}$$

$$SST = 68.4 * (TEX_{86^{H}}) + 38.6 \tag{4}$$





The BIT-Index (Eq. 5) was derived from the relative abundance of branched glycerol dialkyl glycerol tetraethers (brGDGTs) and the isoprenoid GDGT crenarchaeol (Hopmans et al., 2004). While brGDGTs are thought to be produced by soil bacteria, crenarchaeol is known to be a biomarker for planktonic archaea. Hence, the index is widely used as a proxy for soil input (Weijers et al., 2014 and references therein).

$$BIT = \frac{[I+II+III]}{[I+II+III]+[V]} \qquad (5)$$

A relationship between BIT-index and the tetraether index $TEX_{86}^{H}$ (Fig. S2) is not evident, ruling out a possible alteration of the $TEX_{86}^{H}$ signal near big river flows as indicated by Weijers et al., (2014). Comparison of core-top $TEX_{86}^{H}$-based SST estimates from GeoB18308-1 with satellite-derived SSTs (MODIS-A ftp://podaac-ftp.jpl.nasa.gov) reveals similar temperatures (Locarini et al., 2013). Reconstructed SST values are therefore assumed to reflect mean annual values.

**3.5 Grain size**

Particle size distribution was obtained by laser diffraction, using a Malvern Mastersizer 2000 fitted with a Hydro 2000G dispersion unit. Scattered light data was recorded from 2000 to 5000 snapshots of 10 µs. A polydisperse mode of analysis and a refractive index of 1.533 with an adsorption of 0.1 were chosen. Size data collection was performed at constant obscuration in the range 10–20%. Visible shell fragments were removed prior to measurement.

**3.6 Micropaleontology**

Sixteen samples from core GeoB18308-1 were analysed micropaleontologically. The sample volumes vary between 1 and 9 ml around an average of 5 ml and represent 1 cm-thick horizon each. The sediment was sieved with tap water through 63 µm and 200 µm sieves and spilled with aqua dest. into dishes for drying them in an oven at 60°C. The >200 µm size fraction of the residues was picked under a low-power stereomicroscope for all ostracods and foraminifers. Other microfossils or fragments of larger forms were counted on group level and only selected specimens of those were picked as a reference. The 63-200 µm fraction was checked only for taxa not kept by the larger screen. Identification of ostracods relies on Benson and Maddocks (1964), Dingle (1992, 1993, 1994), Dingle and Honigstein (1994) Martens et al. (1996), this of Foraminifera mainly on Lowry (1987) and Schmidt-Sinns (2008). If possible, all taxa were identified to the species level. Light microscope and SEM pictures supported taxonomic work. The material is housed in the collection of PF and SM and will be transferred to the IZIKO Museum in Cape Town, South Africa, after closing of the RAiN project. Abundance data are generated by referring counts to a standardised volume of 1 ml per sample. Relative abundances (percentages) are calculated for samples with at least 20 specimens to keep a large number for statistical analysis; samples contain about 100 specimens on average. Statistical analysis was carried out using the program package PAST (Hammer et al. 2001). A Principal Components Analysis (PCA) was used for identifying main factors structuring the changing association composition within the core (Fig. S3). Association data are composed of relative abundance data grouping selected species on the genus level if ecologically reliable. Only taxa occurring in at least two samples and with a proportion of more than 5% in at least one



sample are considered. Before running the PCA, a Spearman Rank Correlation was applied for identifying highly correlating taxa. None of the considered taxa was found to do so.

### 3.7 Heavy-mineralogy

We have point counted 3 samples and separated heavy mineral from 2 additional samples in the same study area at the "Laboratory for Provenance Studies" of Milano-Bicocca. From a quartered aliquot of each bulk sample (5-10 g), sediments were wet sieved with a standard 500 micron sieve in steel and with a handmade special tissue net sieves of 15 micron. The fraction >500 micron and <15 micron were dried after sieving and weighted for a quantitative estimation of each granulometric class. Heavy minerals were after separated by centrifuging in sodium polytungstate (density 2.90 $g/cm^3$) in the 15-500 micron size-window and recovered by partial freezing with liquid nitrogen. Heavy minerals after separation were also weighted. An appropriate amount of heavy minerals was split and mounted with Canada balsam (n=1.54), and 200 to 250 transparent heavy-mineral grains were point-counted at suitable regular spacing (100 micron) under a polarizing microscope to obtain real volume percentages (Galehouse, 1971). During point counting we have also studied surface textures on detrital grains by polarizing microscope to estimate chemical dissolution (Andò et al 2012). Dubious grains were checked and properly identified by an inVia Renishaw Raman spectrometer equipped with a green laser 532nm and a 50x LWD objective (Andò and Garzanti 2013), in the spectral range (144-4000 $cm^{-1}$) referring to Andò and Garzanti 2013. Heavy-mineral concentration was calculated as the weight percentage of total heavy minerals (HMC) and transparent heavy minerals (tHMC).

### 4 Results

### 4.1 On-and offshore samples

### 4.1.1 Isotope Geochemistry:

Downcore as well as in the catchment samples (Fig. 5; Fig. 6; Table 2; Table 3) the average relative contributions of the long-chain $n$-alkanes are: $n$-$C_{29}$ ~15-23%, $n$-$C_{31}$ ~40-50%, and $n$-$C_{33}$ ~20-24%. Together, these compounds accounted for 80-90% of the total $n$-alkanes. The CPIranged from 7.5-14.4 indicating generally fresh, non-degraded material. The most abundant $n$-alkane is the $n$-$C_{31}$ alkane. Due to the consistent and strong predominance of the $n$-$C_{31}$ alkane further discussion is focused on this compound. $\delta^{13}C$ of the $n$-$C_{31}$ revealed only minor differences (-28.5to-26.7‰ VPDB). The precision is 0.1 ‰ on average with maximum 0.3 ‰. Similarly the Norm31 showed values ranging only minimally from 0.75 to 0.82. Larger differences could be detected in the $\delta D$ of the $n$-$C_{31}$ alkane ranging between (-143 to-127‰). Average precision is 1 ‰ with maximum precision 3 ‰.





### 4.1.2 Heavy-mineralogy

In the studied samples (Fig. 7), heavy-mineral concentrations in silt and sand fraction (15-500m) range from very poor (<0.5 HMC) or poor ($0.5 \leq HMC < 1$) to moderate rich (2 HMC). In the studied sediments heavy minerals display extensive and very common corrosion features on unstable heavy minerals (clinopyroxenes and amphiboles). In Gouritz River catchment sample ECT2-1 clinopyroxenes dominate with subordinate ultrastable zircon, tourmaline, rutile (ZTR), apatite, garnet and titanite, common amphibole and rare orthopyroxene. In the same sample are abundant authigenic crystal of red hematite form beautiful crystal in the shape of flowers on detrital light minerals. The sample, from 11.5 cm depth in Mossel Bay core GeoB18308-1, has a similar heavy-mineral assemblage with lower content of clinopyroxene and a more epidote, very rare chloritoid and orthopyroxene. Red flowers of hematite are detected and corrosion features of clinopyroxenes are similar to the Gourtiz River assemblage. The sample, from 285 cm depth in Mossel Bay core GeoB18308-1 is very similar in composition to the Gouritz River but hematite is extremely rare.

### 4.1.3 Microfossil distribution and PCA results

A diverse foraminifer (at least 46 species) and ostracod (60 species) fauna characteristic for a sublittoral environment was found in the studied core. Beside skeletons of marine invertebrates, continental taxa like plant remains, charophyte oospores, insects, fruits and seeds occur as well, but in low numbers. A continental input is also reflected by six freshwater and four brackish water ostracod taxa occurring in several depths but in rather low numbers. Freshwater ostracods are most abundant in the upper part of the core whereas brackish forms are more typical for this part of the core below about 70 cm. Results of the loading plot analysis of the first principal component shows that charophytes as well as fruits and seeds coming from continental waters or even of terrestrial origin are associated with high PC1 values. Microfossil results can thus be summarized in an index for estuarian inflow (based on PC1 which is best explained by fluvial input) (Fig. S4).

### 4.2 Downcore variations:

Results of AMS-[14]C determination are presented in Table 1. The basal age of core GeoB18308-1 is ~4.100 cal yrs BP and the computed sedimentation rate ranged from 0.1 – 0.6 cm per year, reaching a peak rate between ~1.150-650 cal yrs BP (Fig. 4). The continuous high-resolution, XRF scanning dataset (Fig. S1) indicates little variability in the lower half of core GeoB18308-1. However, a time interval of abrupt increase in Fe, clay and silt content (up to 55%) with more estuarine-inflow-related microfossils (increased PC1), higher BIT-Index (from ~0.06 to 0.81) and enriched δD values of the $n$-$C_{31}$alkane (up to -127‰) can be observed during the period between 1,150 and 650 cal yr BP (Fig. 5). These trends are accompanied by a slight depletion in $\delta^{13}C$ of the $n$-$C_{31}$ alkane (up to -28.5‰) and to generally lower but highly variable $TEX_{86}^{H}$ (from ~0.49 to 0.41) based SST values (12.2 to 17.1°C) (Fig. 5).





## 5 Discussion

### 5.1 Catchment samples-source signatures

#### 5.1.1 Linking catchment depositional processes to rainfall regimes

A relationship between the isotopic composition of rainfall (δD) and the amount of precipitation was described by
Dansgaard, (1964). As hydrogen used for biosynthesis of plant waxes originates directly from the water taken up by the
plants, isotope changes measured in these compounds are related to isotope shifts in precipitation (Sessions et al., 1999).
Catchment samples (all taken at lowland locations) show distinct $\delta D_{C31}$ differences of ~10‰ VSMOW and deuterium
depletion ($\delta D_{C31}$ values ~-138‰) in samples described as paleoflood deposits versus deuterium enrichment ($\delta D_{C31}$ values ~ -
127‰) in samples/horizons from soil layers (see Fig. 6). We attribute this to differences in rainfall regimes contrasting (a) a
deuterium depleted lowland rainfall regime leading to the formation of soils versus (b) rainfall in the highlands which would
be deuterium enriched due to the altitude effect (c.f. Gonfiantini et al. 2001; ca. 10-15 ‰ per 1000m) and lead to flood
events. Meteorological data indicates a mainly most of the area mainly received rainfall during summer when strong easterly
winds provide moisture from a warm Agulhas current Le Maitre et al. (2009). We suggest that plant material synthesized in
the lowlands during this type of climatic setting characterizes the δD signature of soils in the Gouritz catchment. Despite the
dominant summer rains in most of the catchment area, SHW related precipitation events in the otherwise arid winters have
been described as the main precipitation signal influencing the Seweeweekspoort record in the upper most Gouritz River
catchment by Chase et al. (2013). Plant material synthesized in the upper part of the catchment therefore characterizes the δD
signature of paleoflood deposits.

#### 5.1.2 Sediment provenance indicators

##### 5.1.2.1 Leaf wax *n*-alkane distributions and $\delta^{13}C_{C31}$

Different vegetation types show variations in the *n*-alkane distribution of their leaf waxes. In general, it is thought that plants
adapted to aridity produce longer chain wax components than those in habitats of temperate regions (Gagosian and Peltzer,
1986). Therefore, the distribution of *n*-alkane chain length is widely used as an environmental proxy and the ratio *n*-alkane
distribution ratio "Norm31" can indicate changes of sediment source area (Carr et al., 2014). The stable carbon isotopic
composition of organic matter reflects the isotopic composition of the carbon source as well as the discrimination between
$^{12}C$ and $^{13}C$ during biosynthesis (Collister et al., 1994). In particular, compound specific $^{13}C$ of long chained *n*-alkanes
show variations with changes in vegetation type (Collister et al., 1994). In this study, the *n*-$C_{31}$ alkane is the most abundant *n*-
alkane of the plant waxes and thus used as an indicator of vegetation type in addition to Norm31 ratios. Carr et al. (2014)
showed vegetation specific distributions of *n*-alkane homologues within the arid zone South African flora (Fig. 2). In the
case of Gouritz River catchment samples and GeoB18308-1 downcore samples, plant waxes indicate a dominant Karoo
vegetation signature according to Carr et al.(2014) and Hermann et al. (2016) (Fig. 2). This vegetation type is dominant in
the northern parts of the catchment area (see map in Fig. 2). The flood deposits, soil and suspension load samples analyzed in



this study however indicate that even in the lower catchment area the Karoo vegetation signature is dominant in the flood as well as in the soil deposits (Table 2, Fig. 2). This may be attributed to a) the major difficulties in attributing vegetation types to *n*-alkane distributions caused by CAM plants existing in South Africa´s southernmost vegetation (Boom et al., 2014), b) presence of succulent and Karoo plants into the areas classified as "fynbos biome", c) an overprint of the lower catchment

signature by depositional material originating from the upper catchment. Understanding these processes in detail is beyond the scope of this study. Instead, *n*-alkane distributions and their isotopic values are used as provenance indicators. In the Gouritz River catchment as well as downcore samples both indicators of vegetation type have similar signatures (identical average Norm31 values of 0.79 and average $\delta^{13}C_{C31}$ of -28.5‰ and ~-26.7‰respectively) and showed just minor variations (SD of Norm31 and $\delta^{13}C_{C3}$ downcore = 0.02 and 0.4 respectively; n=27 (Fig. 2, Table 2). We can therefore infer that the

sediments deposited at the GeoB18308 site originate directly from the Gouritz River catchment area.

### 5.1.2.2 Heavy-mineralogy

Heavy-mineral analysis represents an independent powerful tool in provenance studies. In the Gouritz River catchment (sample ECT2-1) as well as in both samples from GeoB18308-1 clinopyroxenes dominate with subordinate ultrastable zircon, tourmaline, rutile (ZTR), apatite, garnet and titanite, common amphibole and rare orthopyroxene. The significant

contribution of ultrastable ZTR to the heavy mineral assemblages of GeoB18308-1 samples reflects the Gouritz River Cape Supergroupsandstone-dominated geology. The abundance of corroded clinopyroxene indicates weathering of doleritic dikes located only in the uppermost parts of the Gouritz River catchment. Although mineral and organic loads do not necessarily derive from the same source area, when used as provenance indicators both suggest a (upper) Gouritz River catchment provenance for the sediments deposited at the GeoB18308 site.

**5.2 GeoB18308-1 Paleoclimate record**

**5.2.1 ~4,880-1,150 cal yr BP –stable arid conditions**

The oldest part of the 18308-1 paleorecord (~4880-1150 cal yr BP) is characterized by high Ca, low Fe; mud and silt content below 20%-30% and the particle size dominated by the sand fraction. We associate this with winnowing of fine grained material by the strong current field inherent to the coastal Agulhas Bank. In conditions of low fluvial input the strong current

is liable to remove fluvial derived clay and silt fraction. Heavy mineral analyses from this part of the core (285 cm) show that hematite is extremely rare, suggesting a dry period without formation of iron oxides. Further afield in South Africa, pollen data from Lake Eteza (South African east coast) reported relative dryness for this period. This was deduced from evidence for decreasing trees and shrubs vegetation accompanied by increasing herbaceous plants (Neumann et al., 2010; Scott et al., 2012). The high resolution XRF and grainsize records indicate little variability in this sedimentation regime. This

is in accordance with the relatively stable conditions recorded in the millennia prior to ~1,150 cal yr BP in the speleothem layer width from the Cold Air Cave (Holmgren et al., 1999) as well as pollen data from Wonderkrater and Rietvlei Dam (Scott et al., 2012) in the eastern SRZ.





### 5.2.2 ~1,150-650 cal yrs BP - Medieval Climate Anomaly

At ~1,150 cal yr BP an abrupt shift occurred to more variable (~12 to 17°C), but generally lower SSTs (Fig. 5) in the study area. At the same time, fluvial deposition became more dominant in the record: there is a strong increase in terrigenous input described by a higher index of fluvial input based on microfossil composition, a higher Fe content, an increased BIT

index(from ~0.1 to 0.3) as well as a twofold increase of clay and silt content (to 40-60%) (Fig. 5).This may either indicate a decrease in Agulhas strength, an increase of sediment delivery via the Gouritz River or a combination of both. Simultaneously a shift towards lower $\delta^{13}C_{C31}$ values (to-28.5‰) in the sediment core can be observed. The heavier isotope$^{13}$C is more depleted in $C_3$ plants (as opposed to $C_4$ and CAM plants) which are less adapted to aridity (Collister et al., 1994). Furthermore, $\delta^{13}C$ values may become more depleted when a plant's water-use efficiency decreases in moister

climatic conditions (Pate, 2001; Ehleringer and Cooper, 1988). In either case, $\delta^{13}C$ values of $n$-$C_{31}$ alkanes exported from the catchment suggest a shift towards more humid conditions on land after 1,150 cal yrs BP. At the same time, δD values of the leaf wax $n$-$C_{31}$alkane shift to-129‰ indicating deuterium-enriched precipitation during this time. This likely indicates a shift to source regions with lower altitude precipitation. Enriched $\delta D_{C31}$ values would be an indication for plant material mainly derived from more vegetated lowlands as opposed to plant material derived from the upper catchment during torrential flood

events. Within the error margin of our age-model, the shift we see at ~1,150 cal yr BP towards a more humid lower Gouritz River catchment is a general southern African SRZ trend termed Medieval Climate Anomaly (MCA) summarized by Tyson and Lindesay, (1992) and Tyson and Preston-Whyte (2000)). A large array of continental records document this humid period throughout the SRZ (Talma and Vogel, 1992; Holmgren et al., 1999; Thomas and Shaw, 2002). The few existing records of the YRZ record similar trends; a continuous rise in precipitation was found in $\delta^{15}N$ of hyrax middens at

Seweweekspoort (Chase et al., 2013) and more evidently in the (TOC-based) humidity record at Groenvlei – a Wilderness lake (Wündsch et al., 2016) (Fig. 5). In contrast to the large array of available continental datasets, marine records are rare. The only SST record published for the area (Cohen and Tyson, 1995) does not include data for the time frame in question. However, it does provide a conceptual model of ocean – atmospheric interplay to be tested (c.f. Fig. 8). The authors describe a scenario of a periodically strengthened and more southerly South Indian Ocean Anticyclone (SIA) and South Atlantic

Anticyclone (SAA) having an onshore as well as an offshore effect: 1) reinforcing the tropical easterly influence causing extended warm wet spells in the South African SRZ and 2) increasing the frequency of eastward ridging highs and, thus, along shore winds driving the coastal upwelling on the eastern Agulhas Bank. Additionally, shelf-edge upwelling becomes more frequent when the strong SIA increases the Agulhas volume transport. The decrease in SSTs recorded in GeoB18308-1 for the interval of increased humidity in the Gouritz River catchment inferred in this study for the time interval of the

Medieval Climate Anomaly serves as data to validate this conceptual model.



### 5.2.3 Conditions after ~650 cal yr BP - Little Ice Age and beyond

Continuous sedimentation at the core site was interrupted at ~650 cal yr BP. One or more erosive event deposits are inferred from the sedimentology (erosive contact at ~60-65 cm; fine sand with intercalated organic layers and lumps from ~30-60 cm) and an age reversal indicated by the 2 radiocarbon dates in this interval (~1,466 cal yr BP at a depth of 31 cm and ~640 cal yr BP at a depth of 60 cm; Fig. 4; Table 1). The redeposited material can be characterized as reworked soil material (high BIT-index: ~0.7) supposedly formed in the upper parts of the catchment (the altitude effect depleting the $\delta D_{C31}$ signature to ~-135‰) under humid conditions (according to the depleted $\delta^{13}C_{C31}$ of ~-28‰VPDB) (Fig. 5, Table 3). After ~650 cal yr BP continuous sedimentation is re-established with sediment properties returning to pre-hiatus conditions. The increase in torrential rains and flashfloods that may be inferred for the time period of the event deposit(s) (~300-650 cal yr BP) roughly falls into the timeframe of the so-called "Little Ice Age (LIA)" recorded as humid throughout the South African WRZ (Meadows et al., 1996; Benito et al., 2011; Stager et al., 2011; Weldeab et al., 2013) due to a northward shift of the SHW (Tyson and Preston-Whyte, 2000; Chase and Meadows, 2007). Our catchment study (Leaf wax $\delta D_{C31}$ of paleoflood and soil deposits-see 5.3.1) indicates that paleoflood deposits are primarily induced by an increase in high latitude precipitation. In the uppermost Gouritz catchment (Seweeekspoort site) a major SHW sourced rainfall regime has been documented (Chase et al. 2015). Desmet and Cowling (1999) indicate that despite the general SRZ regime in the Gouritz catchment, the SHW supply additional rainfall in extreme events. The return to "normal flow conditions" recorded at the GeoB18308-1 site after the LIA time interval (i.e. after ~300 cal yr BP) does not indicate any recent major Gouritz River flood events (e.g. the 1983 mega flood documented by Damm et al. in 2010), related to the intensive herding practiced in the catchment between ~ the 1850s and the 1980s (Meadows et al. 1994; Dean and Macdonald, 1994). The absence of an evident anthropogenic signal in the record underlines the magnitude of the LIA events.

### 6 Conclusion

The unique position of the core site directly offshore the Gouritz River mouth at a central location on the South African South Coast allows for a combined analysis of variability in marine processes as well as terrestrial input at a high temporal resolution. In order to reliably reconstruct terrestrial climatic change from an offshore record, measurements were not only performed downcore, but also on material from the Gouritz River catchment for ground-truthing in a source to sink approach. Samples from the river and its flood deposits reveal that the sediments at our core site predominantly originate from the Gouritz River catchment. Furthermore, they indicate that paleoflood deposits are sourced from heavy rainfall events in the highlands whereas soil formation processes are more likely linked to regular rainfalls in the lowlands. Downcore GeoB18308-1 records stable conditions onland and offshore prior to the so-called Medieval Climate Anomaly (~1150-650 cal yrs BP ka), which has been interpreted as humid in the (lowland) region between the eastern SRZ and western WRZ. The Little Ice Age (<~650 cal yr BP ka) interval in turn is characterized by torrential flood events attributed to (SHW) storm tracks influencing the upper catchment area. These paleoclimatic reconstructions correspond to available continental records



for the region. The unique feature of this study at the marine- terrestrial interface is the SST record accompanying the terrestrial climatic reconstructions. This first regional SST dataset for the 2,400-650 yr BP timeframe allows us to test the conceptual model of oceanic – atmospheric interplay (Cohen and Tyson, 1995) during short term Late Holocene climatic anomalies. In accordance with this model (c.f. Fig. 8), our results indicate that variability in South Indian Ocean Anticyclone

(SIA) location and strength drives both tropical easterly influence on South African SRZ climate as well as coastal upwelling on the eastern Agulhas Bank. During a poleward displacement and strengthening of the SIA (e.g. MCA situation) the stronger easterly (alongshore) induce coastal upwelling, common during summer on the eastern Agulhas bank, to become more frequent. In contrast; a weakened, more northerly SIA (e.g. during LIA conditions) has the opposite effect: the weaker Agulhas current is less liable for upwelling and the more frequent SHW advect warm surface water plumes onto the Agulhas

bank in analogy to the modern day winter situation. Future climate change may follow similar patterns and the scenario could be replicated if similar shifts take place with future global warming.

Acknowledgements

This work was financially supported by the Bundesministerium für Bildung und Forschung (BMBF, Bonn, Germany) within

the project "Regional Archives for Integrated Investigation (RAiN)". We also thank the captain, the crew and scientists of the Meteor M102 cruise for facilitating the recovery of the studied material. This study would not have been possible without the support of MARUM. Thanks to all RAiN members for critical discussion and helpful advice for our work progress.

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



**Table 1 AMS radiocarbon analyses of bulk TOC and carbonate material from core GeoB18308-1. Depending on sediment type (terrestrial or marine identified via BIT-Index) the Southern Hemisphere calibration curve (SHCal13) (Hogg et al., 2013) or the modelled ocean average curve (Marine13) (Reimer et al., 2013) were used. To the later a ΔR of 146 ± 85 14C (Dewar et al. 2012) was applied.**

| core | depth (cm) | lab number | 14C age yr BP | material | Cal age yr BP -2s | +2s | median | Curve |
|------|-----------|-----------|--------------|----------|-----|-----|--------|-------|
| GeoB18308-1 | 16.5 | Beta-397268 | -150-±-30-BP | wood | "post-bomb" 98.2 +/-0.4 pMC | | | |
| GeoB18308-1 | 31 | Beta-397269 | -1560-±-30-BP | TOC | 1384 | 1240 | 1466 | SHCal13 |
| GeoB18308-1 | 60 | Poz-61261 | -670-±-30-BP | TOC | 531 | 413 | 640 | SHCal13 |
| GeoB18308-1 | 69 | Beta-397270 | -1370-±-30-BP | TOC | 1196 | 1116 | 1294 | SHCal13 |
| GeoB18308-1 | 123 | Poz-60757 | -1885-±-30-BP | crab claw | 1744 | 1585 | 1834 | Marine13 |
| GeoB18308-1 | 125 | Poz-63976 | -635-±-30-BP | TOC | 665 | 553 | 598 | Marine13 |
| GeoB18308-1 | 146 | Poz-61260 | -1085-±-30-BP | wood | 1057 | 934 | 991 | Marine13 |
| GeoB18308-1 | 195 | Poz-63472 | -1045-±-30-BP | TOC | 1050 | 921 | 953 | Marine13 |
| GeoB18308-1 | 285 | Poz-60754 | -10100-±-50-BP | bivalve | 11829 | 11255 | 11696 | Marine13 |
| GeoB18308-1 | 285 | Poz-63585 | -2075-±-30-BP | TOC | 1984 | 1804 | 2045 | Marine13 |
| GeoB18308-1 | 394 | Poz-63471 | -3020-±-30-BP | TOC | 3194 | 2934 | 3215 | Marine13 |
| GeoB18308-1 | 408 | Poz-63653 | -2910-±-30-BP | bivalve | 3012 | 2814 | 3047 | Marine13 |
| GeoB18308-1 | 440 | Poz-63793 | -3620-±-30-BP | TOC | 3921 | 3697 | 3929 | Marine13 |
| GeoB18308-1 | 490 | Poz-61290 | -4180-±-35-BP | TOC | 4692 | 4439 | 4720 | Marine13 |





**Table 2 n-Alkane isotopic composition and distribution descriptive parameters averaged (with standard deviation).**

| Sample name | latitude (decimal degrees) | longitude | δ¹³C$_{C31}$ ‰ VPDB | standard deviation δ¹³C$_{C31}$ | δD$_{C31}$ ‰ VSMOW | standard deviation δD$_{C31}$ | Norm31 | sample material |
|---|---|---|---|---|---|---|---|---|
| ECT-1-2 | -34,28081 | 21,82842 | -26,77 | 0,05 | -127,29 | 2,69 | 0,85 | soil horizon |
| ECT-1-3 | -34,28081 | 21,82842 | -29,10 | 0,10 | -136,96 | 0,43 | 0,83 | paleoflood deposit |
| ECT-2-1 | -34,18470 | 21,75288 | -28,84 | 0,05 | -127,30 | 0,22 | 0,76 | paleoflood deposit |
| ECT-3-1a | -34,08183 | 21,74178 | -27,92 | 0,21 | -129,86 | 0,92 | 0,78 | soil horizon |
| ECT-3-1b | -34,08183 | 21,74178 | -27,74 | 0,07 | -139,18 | 0,83 | 0,73 | paleoflood deposit |
| ECT-5-1 | -33,90890 | 21,65330 | -29,50 | 0,08 | -130,13 | 0,13 | 0,78 | soil horizon |
| ECT-5-2 | -33,90890 | 21,65330 | -29,44 | 0,15 | -132,31 | 0,23 | 0,81 | paleoflood deposit |
| ECT-6-1 | -33,90917 | 21,65353 | -29,14 | 0,05 | -134,38 | 1,09 | 0,83 | paleoflood deposit |
| ECT-6-2 | -33,90917 | 21,65353 | -28,14 | 0,10 | -128,87 | 0,09 | 0,82 | soil horizon |
| ECT-7-2 | -33,75837 | 21,46945 | -29,66 | 0,08 | -131,94 | 1,35 | 0,74 | riverbank deposit |
| ECT-8-1 | -33,62422 | 22,22843 | -31,56 | 0,06 | -128,20 | 0,23 | 0,77 | riverbank deposit |





**Table 3 GeoB18308-1 organic geochemical downcore data. *n*-Alkane isotopic composition and distribution descriptive parameters averaged (with standard deviation), BIT-Index, TEX$_{86}^{L}$ and SSTs(°C) calculated from TEX$_{86}^{H}$ after Kim et al. (2010).**

| Depth (cm) | $\delta^{13}C_{C31}$ ‰ VPDB | standard deviation $\delta^{13}C_{C31}$ | $\delta D_{C31}$ ‰ VSMOW | standard deviation $\delta D_{C31}$ | Norm31 | BIT | TEX$_{86}^{H}$ | SST(°C) (Kim et al. 2010) |
|---|---|---|---|---|---|---|---|---|
| 2,5 | -28,490 | 0,284 | -134,312 | 1,978 | 0,802 | 0,211 | 0,477 | 16,6 |
| 12,5 | -28,480 | 0,049 | -130,655 | 0,267 | 0,790 | 0,301 | 0,483 | 17,0 |
| 17,5 | -27,630 | 0,045 | -133,604 | 0,653 | 0,816 | 0,701 | 0,538 | 20,2 |
| 32,5 | -27,969 | 0,168 | -136,375 | 1,441 | 0,814 | 0,813 | 0,593 | 23,1 |
| 52,5 | -27,802 | 0,009 | -135,387 | 1,719 | 0,820 | 0,526 | 0,538 | 20,2 |
| 67,5 | -28,063 | 0,003 | -133,719 | 0,800 | 0,790 | 0,732 | 0,623 | 24,5 |
| 77,5 | -28,286 | 0,026 | -129,138 | 2,394 | 0,777 | 0,098 | 0,477 | 16,6 |
| 107,5 | -27,724 | 0,006 | -136,198 | 0,822 | 0,789 | 0,132 | 0,484 | 17,1 |
| 112,5 | -27,446 | 0,118 | -133,986 | 0,155 | 0,801 | 0,306 | 0,481 | 16,9 |
| 115 | -27,449 | 0,004 | -126,742 | 0,999 | 0,824 | 0,193 | 0,488 | 17,3 |
| 122,5 | -27,781 | 0,295 | -131,669 | 0,458 | 0,806 | 0,094 | 0,409 | 12,0 |
| 130 | -27,714 | 0,040 | -127,357 | 0,338 | 0,793 | 0,274 | 0,447 | 14,7 |
| 142,5 | -27,757 | 0,175 | -131,973 | 0,225 | 0,800 | 0,194 | 0,492 | 17,5 |
| 152,5 | -27,696 | 0,089 | -134,566 | 1,495 | 0,775 | 0,101 | 0,482 | 16,9 |
| 155 | -27,461 | 0,054 | -133,870 | 0,728 | 0,782 | 0,224 | 0,444 | 14,5 |
| 162,5 | -27,871 | 0,046 | -136,287 | 0,535 | 0,768 | 0,343 | 0,524 | 19,4 |
| 165 | -26,706 | No value | -130,842 | 1,740 | 0,775 | 0,229 | 0,448 | 14,8 |
| 172,5 | -27,323 | 0,161 | -133,485 | 0,921 | 0,770 | 0,156 | 0,462 | 15,7 |
| 180 | -27,454 | 0,001 | -135,430 | 0,911 | 0,775 | 0,313 | 0,456 | 15,2 |
| 187,5 | -27,633 | 0,149 | -136,050 | 0,655 | 0,762 | 0,309 | 0,496 | 17,8 |
| 207,5 | -27,430 | 0,222 | -136,687 | No value | 0,785 | 0,063 | 0,477 | 16,6 |
| 287,5 | -27,459 | No value | -137,911 | 2,688 | 0,803 | 0,126 | 0,460 | 15,6 |
| 327,5 | -27,475 | 0,072 | -139,901 | 1,058 | 0,769 | 0,117 | 0,474 | 16,4 |
| 397,5 | -27,213 | 0,027 | -141,268 | 0,301 | 0,765 | 0,150 | 0,463 | 15,7 |
| 452,5 | -27,610 | 0,023 | -138,820 | 1,432 | 0,781 | 0,159 | 0,471 | 16,2 |
| 482,5 | -27,626 | No value | -142,785 | No value | 0,754 | 0,100 | 0,471 | 16,3 |
| 487,5 | -27,181 | 0,023 | -137,367 | 0,767 | 0,749 | 0,087 | 0,471 | 16,3 |



**Table 4 Mineralogical composition of heavy minerals in silt and sand fraction (15-500 m) of marine sediments from GeoB18308-1 and Gouritz River catchment**

| location | sample name | HM %weight | zircon | tourmaline | rutile | titanite | apatite | epidote | garnet | chloritoid | sillimanite | amphibole | clinopyroxene | orthopyroxene | Total | ZTR | % transparent | % opaque | % Fe oxides | % Ti oxides | % turbid HM | **Total** |
|---|---|---|---|---|---|---|---|---|---|---|---|---|---|---|---|---|---|---|---|---|---|---|
| Gouritz River: Bland´s Drift | ECT2-1 | 1,5 | 4 | 2 | 1 | 5 | 6 | 9 | 7 | 0 | 0 | 4 | 62 | 1 | **100** | 6 | 67% | 8% | 15% | 9% | 1% | **100%** |
| Mossel Bay 5km offshore | GeoB18308-1 (11cm) | 1,3 | 6 | 4 | 5 | 3 | 6 | 19 | 5 | 1 | 0 | 5 | 46 | 2 | **100** | 15 | 72% | 10% | 7% | 11% | 0% | **100%** |
| Mossel Bay 5km offshore | GeoB18308-1 (285 cm) | 1,3 | 7 | 2 | 2 | 3 | 4 | 10 | 7 | 0 | 0 | 2 | 60 | 2 | **100** | 12 | 77% | 10% | 1% | 12% | 0% | **100%** |



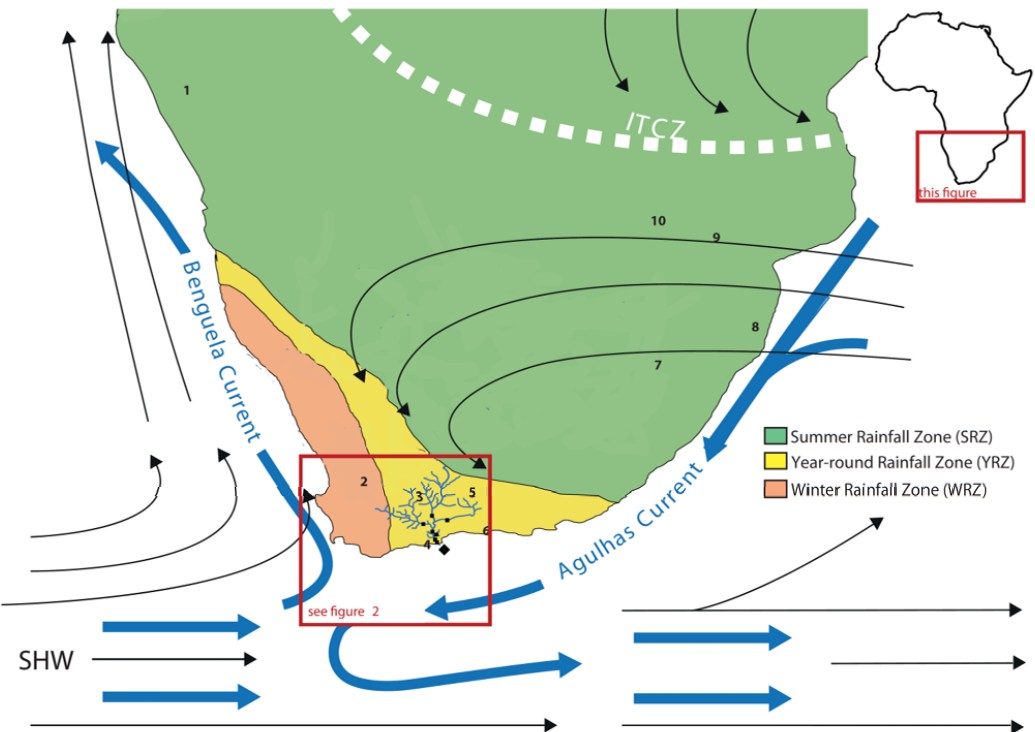

**Fig. 1 Schematic overview of the main components of the oceanic (thick arrows) and atmospheric (thin arrows) circulation over southern Africa (modified from Truc et al., 2013). The Winter Rainfall Zone (WRZ) is colored in red; the Summer Rainfall Zone (SRZ) in green and the year-round rainfall zone (YRZ) in yellow. Circles indicate sampling positions; core GeoB18308-1 marked in red. SHW = Southern Hemispheric Westerlies; CAB = Congo Air Boundary; ITCZ = Intertropical Convergence Zone. Note**
5   **that these are shown in their summer position. Austerlitz midden (1), Katbakkies Pass midden (2), Seweweekspoort midden(3), Pinnacle Point (4), Cango Cave (5), Nelson Bay Cave (6), Braamhoek peat (7), Lake Eteza (8), Cold Air Cave (9), Wonderkrate (10).**





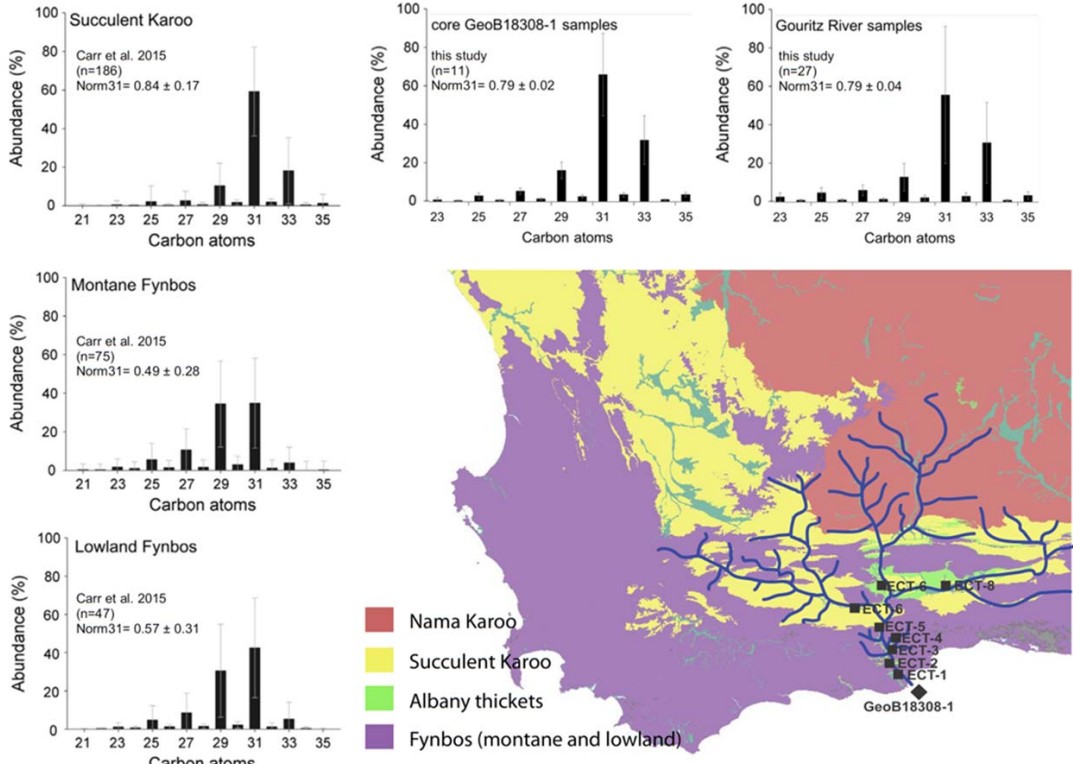

**Fig. 2 The map is an inlet of Fig. 1 and shows vegetation types in the drainage basin (after Mucina and Rutherford, 2006 and Scott et al., 2012) of the Gouritz River (blue). Squares indicate localities of Gouritz River samples (ECT= East Coast Trip); diamond indicates marine sediment core location. Plots showing regional biome/eco-region soil and vegetation average *n*-alkane distributions from Carr et al. 2014 are indicated on the left whereas average *n*-alkane distributions of riverine and marine samples from this study are plotted above the map.**





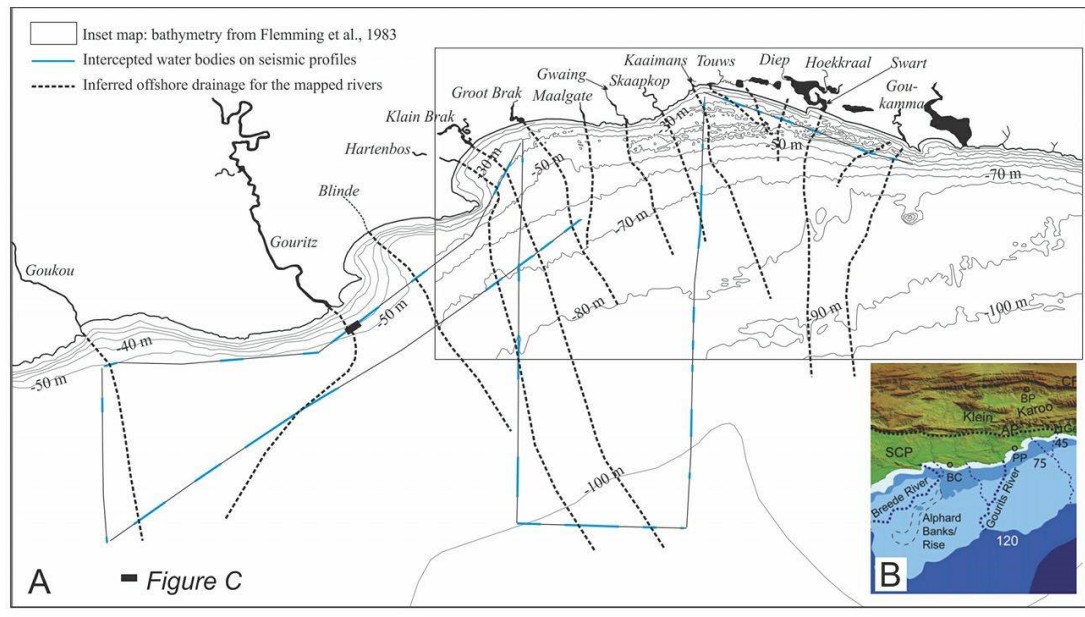

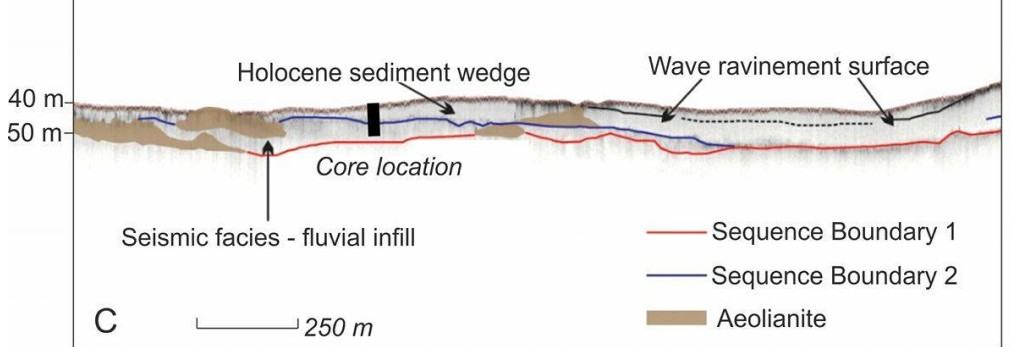

**Fig. 3 (A) Palaeodrainage of the South Coast, prior to infilling and subsequent burial of the channels to form the smooth surface of the continental shelf. From Cawthra (2014). Bathymetry is derived from Flemming et al., 1983 and regional GEBCO datasets. (B) Inset map shows inferred drainage on the shelf from Compton (2011). (C) Pinger seismic profile showing the context of the core.**





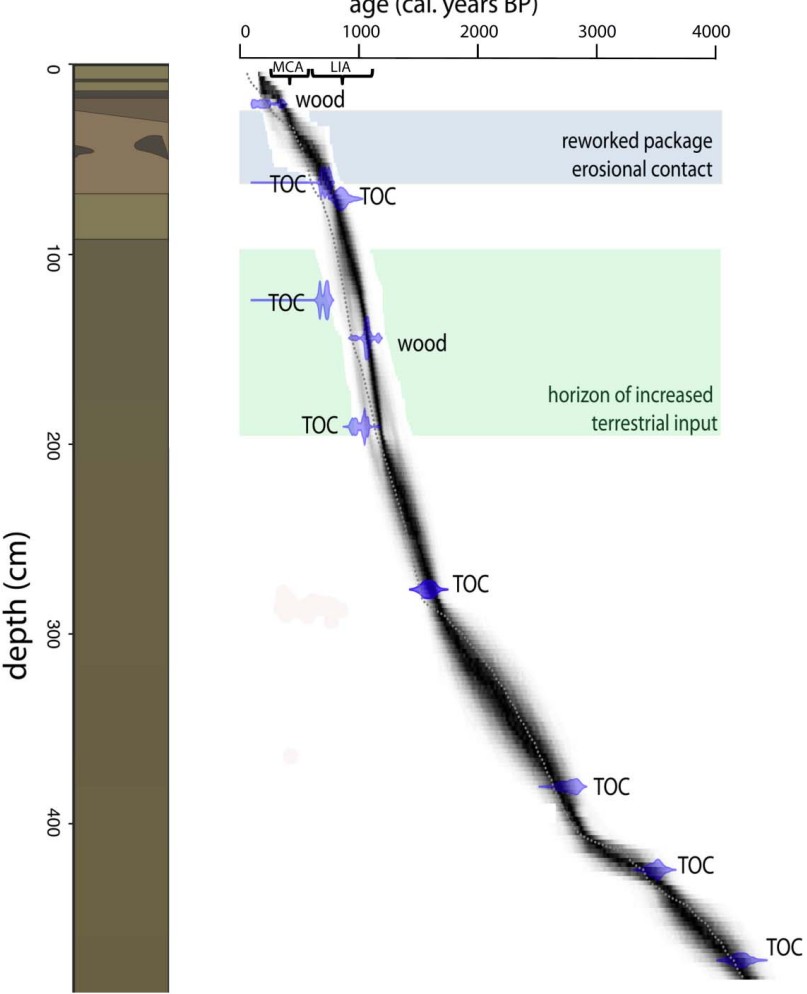

Fig. 4 on left: core log of GeoB18308-1 (internal structures with recorded colors) demonstrating the presence of an event deposit (blue box). On right: age-depth model with distributions of all 10,000 calculated iterations shown as greyscales at 500 equidistant depths. The green box marks horizons of increased terrestrial input where a terrestrial calibration curve (SHCal13: Southern
5   Hemisphere (Hogg et al., 2013)) was used. Otherwise, Marine13-modelled ocean average (Reimer et al., 2013) with a δR following Dewar et al. (2012) and Meadows et al. (in prep/personal communication). Redeposited material (i.e. a crab claw at 123 cm depth, a bivalve at 285 cm and the reworked package at 26–66 cm depth)) were excluded from the age-depth modeling. Medieval Climate Anomaly (MCA) and the Little Ice Age (LIA) are indicated.





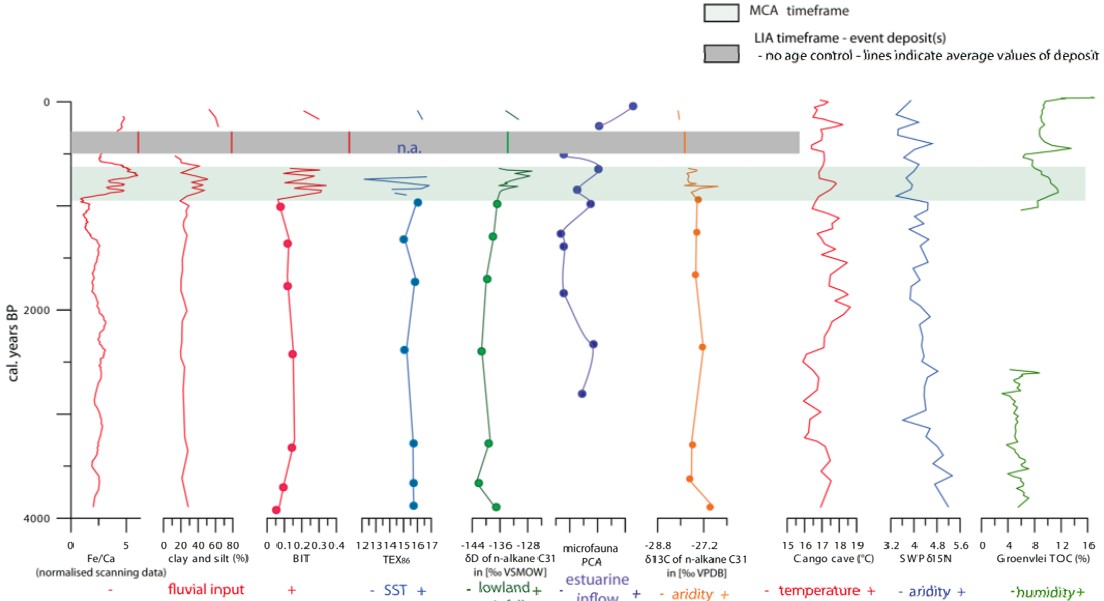

**Fig. 5 organic and inorganic downcore geochemistry of GeoB18308-1 as indicators for fluvial input, SST, SRZ and aridity. Low resolution records are marked by colored dots. SSTs for samples with BIT index above 0.3 were not calculated. The indicator of estuarian inflow is deduced from factor loadings along axes 1 of a PCA on microfossil distribution of core samples. For comparative purposes regional paleoenvironmental records are plotted: paleo-temperature from Cango Cave (Talma and Vogel, 1992), aridity index from Seweeekspoort (Chase et al., 2013) and humidity index from Groenvlei (Wündsch et al., 2016). Shaded areas refer to the time interval of the Medieval Climate Anomaly (dark grey) and the Little Ice Age redepositional event (light grey).**





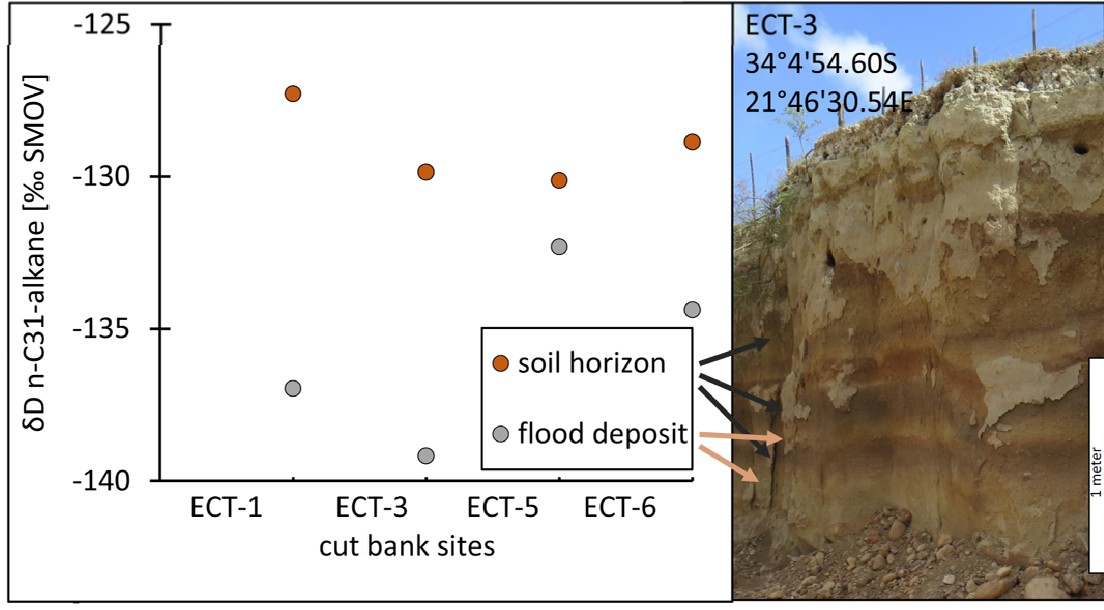

Fig. 6 variations in δD of the *n*-C31-alkane [‰VSMOW] in the distinct horizons of paleoflood vs. soil formation horizons. Gouritz River tributary cut bank at sample location ECT-3 is depicted as an example illustrating these alternating horizons.



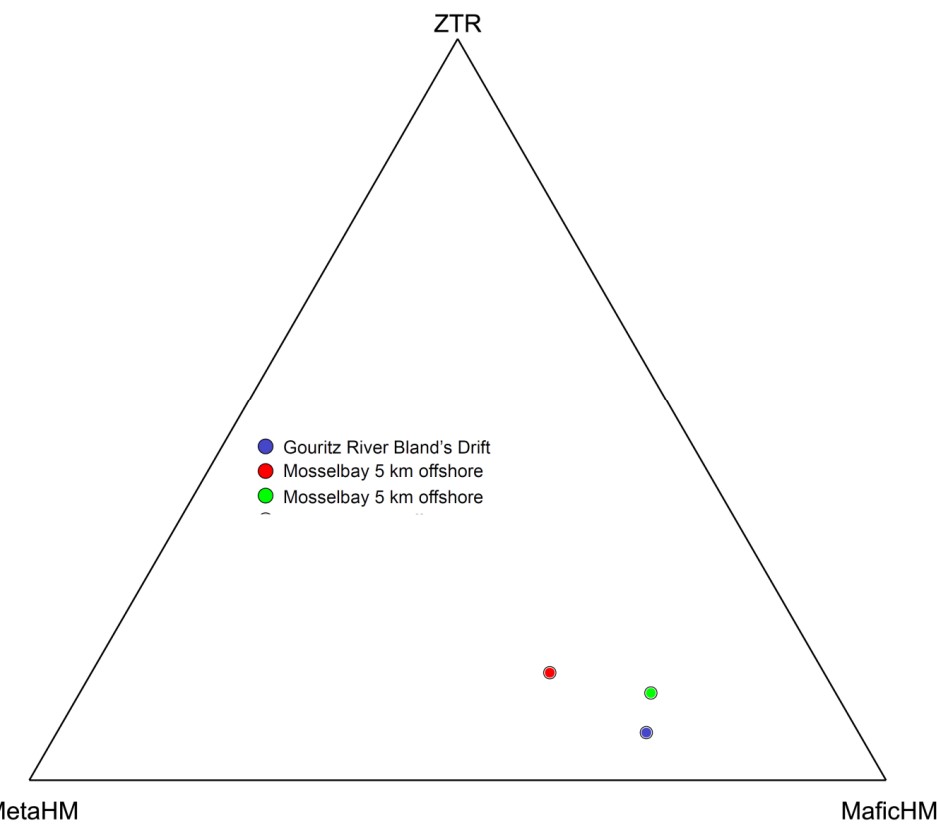

Fig. 7 heavy mineral suite in silt and sand fraction (15-500 m) collected in Mossel Bay plotted according to their ZTR (zircon-tourmaline-rutile); MetaHM (titanite, epidote, garnet, chloritoid, sillimanite, amphibole): MaficHM (apatite, clinopyroxene, orthopyroxene) content.





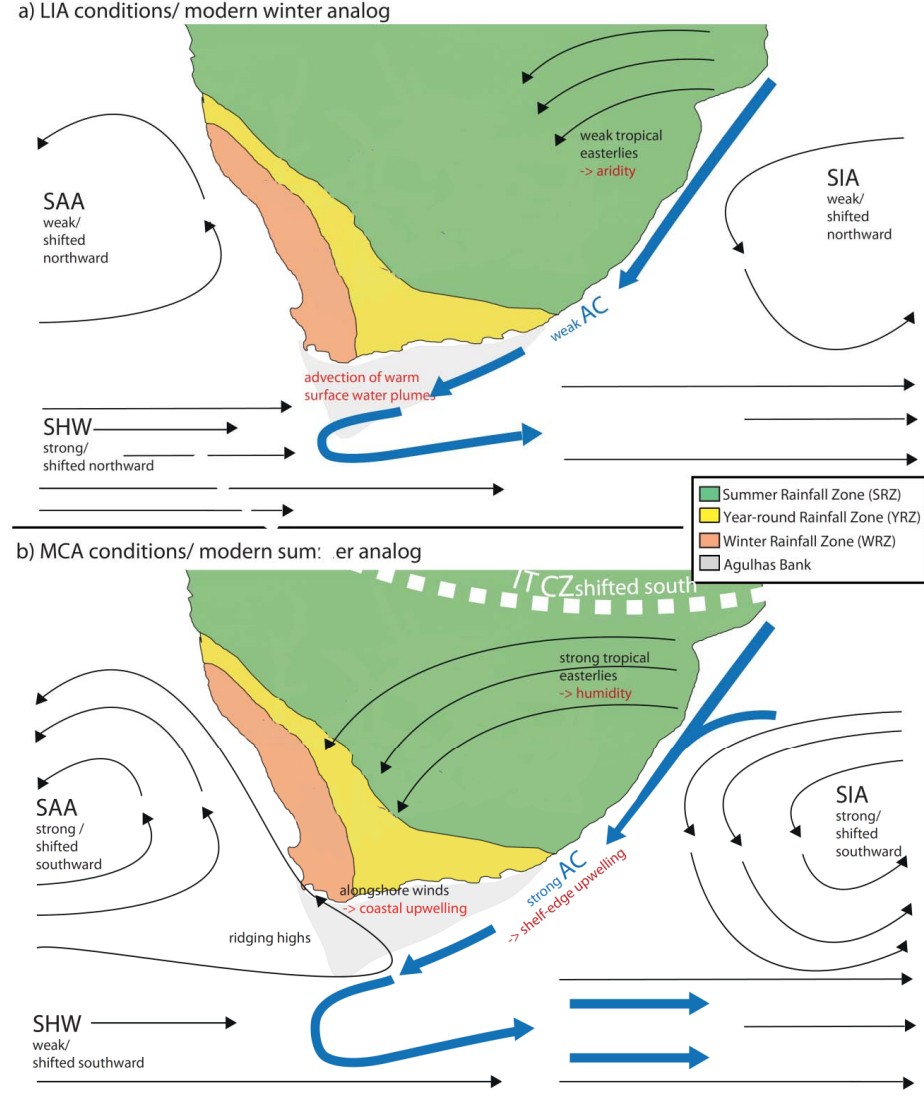

**Fig. 8 Conceptual model after Cohen and Tyson (1995) depicting the recorded response (in red) to shifts in atmospheric (indicated in black) and oceanic (in blue) circulation patterns during the LIA (a) and MCA (b). SIA (=South Indian Anticyclone), SAA (=South Atlantic Anticyclone), SHW (=Southern Hemispheric Westerlies), AC (=Agulhas Current), LIA (=Little Ice Age), MCA (=Medieval Climate Anomaly), ITCZ = Intertropical Convergence Zone.**