# Peer review of "Linking catchment hydrology and ocean circulation in Late Holocene southernmost Africa"

_Climate of the Past, 2016_

## Short Comment (SC1) · 13 Nov 2016

Sebastian Luening

luening@uni-bremen.de

This paper thoroughly documents an important new offshore palaeoclimate datapoint for southern South Africa. The results are based on sophisticated analytical methods and are well explained so that also non-specialists can understand their climatic signficance.

One minor point is the age of the onset of the Medieval Climate Anomaly (MCA) in this dataset. In Chapter 5.2.2 the authors describe the onset of the MCA at 1,150 yrs BP (corresponding to 800 AD). In Fig. 5, however, the MCA is marked as beginning at 950 yrs BP (1000 AD), which fits with the onset of increased fluvial activity and the upwelling-related temperature drop. The MCA start date in the text may need to be changed.

[Figure]

I really like the MCA palaeoclimate map in Fig. 8. This illustrates the processes in a transparent and effective way. The authors present suitable temperature and hydroclimate data by Chase et al. 2013, Talma Vogel 1992 and Wündsch et al. 2016. It is noteworthy that a number of other studies in southern Africa support the trend towards more humidity during the MCA. The studies are marked by green (=more humid) dots on the following online map of an African-wide MCA literature-mapping project:

http://t1p.de/mwp

https://www.researchgate.net/project/Mapping-the-Medieval-Climate-Anomaly

Other South Africa data points with wet MCA:

Scott et al. 2005: Blydefontein basin, Kikvorsberge

Brook et al 2015: Wonderwerk Cave

Neumann et al 2014: Core M11, Mahwaqa Mountain, KwaZulu-Natal

Norström et al 2009: Core AH1, Braamhoek Wetland, Free State

Stager et al 2013, Neumann et al. 2008: Lake Sibaya, KwaZulu-Natal

Woodborne et al 2015: Pafuri area, northern South Africa

Mozambique:

Ekblom Stabell 2008: Lake Nhaucati
* * *

---

## Short Comment (SC2) · 14 Nov 2016

Thank you for your interest in our research! The link and additional citations you have forwarded are helpful additions in supporting our findings on South African MCA climate. Indeed our data (Fig. 5) indicate an onset at 950 yrs BP (1000 AD). We will be sure to modify the text accordingly during the revision process.

---

## Referee Comment (RC1) · Anonymous Referee #1 · 12 Dec 2016

This study represent a multi-proxy approach based on a marine sediment core GeoB18308-1, located on the South Coast of South Africa, offshore the Gouritz River, Southern Cape. The study reconstructs approximately the last 4 ka and additionally presents samples inland within the catchment areas of the Gouritz River itself. The authors interpret their data as demonstrating humid conditions in the Gouritz River catchment during the Medieval Climate Anomaly with lower, but highly variable sea surface temperatures in the Mossel Bay area. On the contrary they claim that the Little Ice Age was characterized by relatively warm sea surface temperatures in Mossel Bay and arid climatic conditions favorable to torrential flood events sourced in the Gouritz headlands.

I am generally excited about this work as it shows new data from an area missing detailed marine/terrestrial records. I think it is a very detailed and solid approach particularly having material from the Gouritz River catchment for "ground-truthing" in a source to sink approach. I am also okay with the conceptional model explaining the atmospheric circulation system which is based on A.L. Cohen and P.D. Tyson 1995.

I am in favour of publication of this record, however would like the authors to respond/check some aspects of the paper which I describe below. My main two concerns are the construction of the age model and the interpretation of the data in the LIA.

Age Model

First of all, if two labs are used, Poznan and Beta lab in this case, it should be shown that the results are consistent between labs. Has a comparison on an aliquot sample been done which shows that both labs come to the same conclusion? Table 1 shows all material dated but only Figure 4 caption reveals what was used for the age model thereafter. It should be clarified in table 1, which of the core depths were not part of the age model. As I understand depths 123 cm, 285 cm and the reworked package at 26-66 cm depth were excluded from the age model. Hence the levels taken into account are 16.5 cm, 69 cm, 125 cm etc. Core depth 69 cm, TOC measured, gives a calibrated age (median) according to table 1 of 1294 cal. Age BP. The level used thereafter is 125 cm, TOC measured, gives a median of 598 cal. Age BP. Also the two levels below are significantly younger than core depth 69 cm dated. Why were these samples part of the age model and not excluded although they could be equally reworked material? In fact the TOC sample at 125 cm just plots outside the uncertainty level given by the Baysian age model. Normally this software should give a probability estimate stating how likely this date is part of the age model or not. I somehow also see a mismatch between the table 1 data and Fig. 4. For example the plot shows two TOC point just below the blue shaded area 'reworked package, erosional contact'. I believe that this refers to the sample at 60 and 69 cm core depth according to table 1. However, the author writes that samples between 26-66 cm were excluded. So the sample at 60 cm should not be in there. Moreover, 490 cm core depth has a median age of 4720 cal. Age yr BP which is not even part of the axis in Fig. 4. And there are more examples

were the cal. age from table 1 does not fit the cal. age on the axis of Fig. 4.

If the author could clarify this mismatch and the core depths used and revise.

It is not clear to me why one would calibrate with an SHcal and then with the marine 13 in the core intervals below despite high BIT index in that interval? Moreover, the BIT index wasn't even measured on the same samples the TOC was dated. Why was there no radiocarbon dating conducted on foraminifera from the same material?

Interpretation of the LIA interval:

I am not sure the data during LIA supports the claim made for that interval. The author states that there is missing age control in that period due to re-depositional events. So no data is shown. Instead the author concludes that based on redeposited material which can be characterized as reworked soil, the time frame of the LIA must have had torrential rains and flashfloods on the background of an arid climate. I can't see the evidence for that conclusion neither for the claim, that the SST's in Mossel Bay were warm if there is no TEX data for that core depth presented. Moreover, what does the average line for deposit mean for the interval shown in Fig. 5? The SST conclusion in this paper is with odds of Zinke et al., 2014 (Zinke, J., B. R. Loveday, C. J. C. Reason, W. C. Dullo, and D. Kroon (2014), Madagascar corals track sea surface temperature variability in the Agulhas Current core region over the past 334 years, Sci. Rep., 4. doi: 10.1038/srep04393) who show that Agulhas Current SSTs cooled through the Little Ice Age.

How can these opposing findings be explained? Moreover, I think there should be more evidence for that claim presented.

2) Specific questions/issues:

Page 2 line 4: I feel that a more African specific chapter of the IPCC report should be cited here rather than Metz or Kirtman et al :

"Niang, I., O.C. Ruppel, M.A. Abdrabo, A. Essel, C. Lennard, J. Padgham, and P.

Urquhart, 2014: Africa. In: Climate Change 2014: Impacts, Adaptation, and Vulnerability. Part B: Regional Aspects. Contribution of Working Group II to the Fifth Assessment Report of the Intergovernmental Panel on Climate Change [Barros, V.R., C.B. Field, D.J. Dokken, M.D. Mastrandrea, K.J. Mach, T.E. Bilir, M. Chatterjee, K.L. Ebi, Y.O. Estrada, R.C. Genova, B. Girma, E.S. Kissel, A.N. Levy, S. MacCracken, P.R. Mastrandrea, and L.L. White (eds.)]. Cambridge University Press, Cambridge, United Kingdom and New York, NY, USA, pp. 1199-1265.

Page 2 line 20: There are more recent studies by now showing insolation driven responds of Southern Africa climate and should be cited here: (Daniau, A.-L., M. F. Sánchez Goñi, P. Martinez, D. H. Urrego, V. Bout-Roumazeilles, S. Desprat, and J. R. Marlon (2013), Orbital-scale climate forcing of grassland burning in southern Africa, Proceedings of the National Academy of Sciences, 110(13), 5069-5073. doi: 10.1073/pnas.1214292110); (Simon, M. H., M. Ziegler, J. Bosmans, S. Barker, C. J. C. Reason, and I. R. Hall (2015), Eastern South African hydroclimate over the past 270,000 years, Scientific Reports, 5, 18153. doi: 10.1038/srep18153)

Page 3 Line 1: Biastoch et al., 2009a does not show that strong SHW reduce leakage into the SA and should not be cited here in this respect. This study only shows what effect shifting the SHW to Leakage strength has. It does not evaluate what a change in the strength of the SHW does to leakage variability.

In this respect the citation of Durgadoo et al., 2013 in the line below is wrong as in this paper the authors show that an equatorward shift in westerlies increases leakage and not like written in this paper page 3 line 4:" a weakening of the Agulhas Current and the leakage of warm water due to northward displacement of the SHW".

Page 5 line 21: Not sure how this description of the bathymetry fits into this part of the oceanography. Would suggest shifting that.

Page 6 line 29: Is that a valid common method to calibrate XFR scans? I would rather think that taking sub-samples and analyzing for bulk major and trace elements would be

the way to do it? One approach could be following the below: "Prediction of Geochemical Composition from XRF Core Scanner Data: A New Multivariate Approach Including Automatic Selection of Calibration Samples and Quantification of Uncertainties By G. J. Weltje, M. R. Bloemsma, R. Tjallingii, D. Heslop, U. Röhl and Ian W. Croudace."

Page 8 line 19: Why and how was the original method modified? Does the modification have advantages compared to Hopmans protocol? If so that should be stated there.

Page 9 line 27: To be statically significant one have to at least count 150 specimens per sample not only 20.

Page 12: By which evidence sedimentological etc. was a paleosoil and a flood deposit distinguished? Only by different dD values? That should be better described and presented in the text. Page 12 line 10: why would rainfall in the highlands automatically lead to flood events? Page 14 line 15: If that is stated then values should be given as well. As the age model was derived from a Baysian approach one can give an uncertainty value here for the age model. Page 14 line 17: The recent review paper by Nash, D. J., G. De Cort, B. M. Chase, D. Verschuren, S. E. Nicholson, T. M. Shanahan, A. Asrat, A.-M. Lézine, and S. W. Grab (2016), African hydroclimatic variability during the last 2000 years, Quaternary Science Reviews, 154, 1-22. doi: http://dx.doi.org/10.1016/j.quascirev.2016.10.012 Should be included in that part of the manuscript.

Also Woodborne, S., G. Hall, I. Robertson, A. Patrut, M. Rouault, N. J. Loader, and M. Hofmeyr (2015), A 1000-Year Carbon Isotope Rainfall Proxy Record from South African Baobab Trees (*Adansonia digitata* L.), PLoS ONE, 10(5), e0124202. doi: 10.1371/journal.pone.0124202 should be added here as their record also shows the wettest period was c. AD 1075 in the Medieval Warm Period.

Fig. 5: For comparative purposes other regional paleoenvironmental records are plotted. How was secured that there are no age model offsets between this study and the other records?

3) Technical corrections

Fig. 2: page 29 line 5: legend says Carr et al., 2014 in the figures in the map it says Carr et al, 2015

Fig. 5 Could do with more labels on the Y-Axis i.e. at least 500 year tick labels between tick marks. Page 12 line 12: twice 'mainly used here! Rephrase grammar is wrong in that part of the sentence! Page 12 line 26: Formatting issues and missing space.

Page 13 line 21. Fig. 5 shows the main record only till 4 ka according to the axis however the text states: " The oldest part of the 18308-1 paleorecord (∼4880-1150 cal yr BP)......where is the rest of the data?

Where are the figure captions of the supplement? And what are the dots in SF1? The calibration samples or the subsampling for the organic geochemistry?

---

## Referee Comment (RC2) · Anonymous Referee #2 · 20 Dec 2016

The study encompasses an impressive variety of methodologies and of different proxies, and discusses the rich results in a convincing way, especially for what regards the aridity-humidity multi-proxy reconstruction. An undoubtable strength of the approach is the analysis carried out in samples from several places in the Gouritz catchment, which provides decisive supports for the inferences made. I would recommend that the manuscript be published, although prior to that the authors should improve a few aspects of it, and respond to some questions that I detail below. General reservations that I have with this work are: 1) a better effort could be made of emphasizing, especially in the abstract and introduction (and potentially also in the title), what the key findings are and what their importance is. As it is it resembles more an account of analyses carried out in a very good setting (whose importance could be made even more clear). 2) It would be very interesting if the authors could draw more explicitly the

implications of their results, and/or of the conceptual model they somehow validate, for the latitudinal shifts in ITCZ. This is of interest to a larger climatological community, and to projections of what the region may expect with ongoing climate change. 3) the paper is very wordy, especially in its sections 2 to 5. The authors should improve readability and really consider refraining from reporting all they have done and all results, and focus of what is of relevance to the new findings discussed. Some records barely matter for the discussion. 4) Probably because of the vast amount of material presented, the manuscript is sloppy in many parts: odd sentences, mismatches in the wording, punctuation, typos. One would expect that nine authors could proofread the manuscript to a higher quality.

Main specific points

* The first sentence of the introduction seems inconsequential and unjustified to me: there is no argument for the importance of South Africa's geographic position. Rephrase. * I would suggest to pay more attention to streamlining the introduction chapter: as it is it is hard to read, and the main points that the authors wish to make do not come through clearly. What are the main research gaps regarding South African climate? Can you present the evidence for one or the other explanation in a more organized manner? * I would recommend an effort to focus section 2. It could be made more concise, and thus the readability of the paper could improve, if you privilege the information that is relevant to the findings of this paper. E.g., the reader doesn't gain insight that are relevant to this Late Holocene paper by your discussing Cretaceous tectonics. * Fig 4. The LIA follows the MCA, not the other way around. * Pag 14 line 2 and following. First, from fig 5 one would say that all discussed changes happen from ca. 950 yBP, rather than 1150. Can you clarify whether the figure or the discussion are correct? Further, I don't think you can state that anything happens to the SST record around 1150 yBP, at least from the results contained in fig.5. Simply the sampling temporal resolution increases, but I would argue there is no real difference in variability before and after 1150 kBP. If anything, low peaks appear after that time: could you

show a real statistical significance between the average SSTs either side of 1150kBP? * (related to one of the main objections of reviewer 1) You state that age-reversals occur from 650 yBP, but from figure 5 one can see that you take data up to ca. 500 yBP seriously (also the gray band starts at 500): can you clarify? * Pag 14 ll 27-28. For intensification of Agulhas Current transport, you should check Durgadoo et al (2013), who report, from three ocean models, that northward shifts (and intensification) in atmospheric features increase Agulhas Current transport contrary to what included in the conceptual model here discussed. This does not mean that ocean models in the above studies hold the truth, but I would suggest you could take the occasion to discuss this contradiction in the literature. (also in the Conclusions) * Pag 16 line 10. Future climate change may follow what pattern? You reported two. Also, could you provide a reference supporting this? (rephrase anyway, as sentence is confused)

Minor comments:

* The title could be modified to eliminate the present continuous tense – vague – and include any word that reports the results of this "linking" * Abstract: "highly dynamic" and "highly complex" used just one sentence apart, maybe either make more specific or eliminate one. "give information on climatic changes": it is vague, make more concrete. The last sentence is unclear: to which processes do you refer, to those in the LIA or in the MCA? Rephrase. Also, probably not appropriate to only refer to a climate model like this at the end of the abstract, where the reader cannot make much out of it: essential information is missing. * Ll 9-12. This sentence is complicated and doesn't show a contrast between concepts that one would expect from the use of "while". * Ll 16 ITCZ not explained, maybe avoid abbreviation as never used anymore. * Ll 16-18 you either use whether, or add a question mark, not both. * Page 3 line 4. Durgadoo et al 2013 find precisely the opposite, i.e. that Agulhas leakage increases when westerlies move north. I would suggest you deal with this in the introduction. * line 6. What is YRZ? * Ll 16-17. Odd phrasing, a sediment core doesn't aim to anything. * Line 28. Harmonize the units (use exponential in place of Mm3) * Line 32. What do you mean

by mixed summer and winter rainfall? And why this single paleo piece of information in a present-day context? * Page 6 ll 15-16. Sentence not clear: what is the unit for the numbers in parenthesis, years? (14C should have 14 in the superscript) * Dewar et al 2012 is missing from the ref list. * Line 17. Why do you inform about the sedimentation rate: this is not further discussed in the paper. * Line 21. Analyses. * Line 24. Change "elemental profiles" in place of "scanning data". * Line 27. Change "vertical resolution or downcore resolution" in place of depth resolution. What are 1.2 cm2? * Line 3. Change "scanning intensities" for a more appropriate term * Pag 7 ll 4-5. Not clear how the xrf data helped in selecting the samples for organic geochemistry, please reformulate. * Methods: try to avoid so many abbreviations, especially those not further used in the paper. In general the manuscript is highly packed with abbreviations, please try to be parsimonious with them. * Pag 9 line 16. "micropaleontologically" probably not a word. * Pag 10 line 2. "None of the considered taxa was found to correlate". * Line 4. What do you mean by point counted? * Line 15. Same reference occurs twice. * Line 20. Fig S4 does not exist. * Pag 14 ll 15-17. The reader gets the impression that the MCA is a Southern African phenomenon, while this is a concept normally applied north Atlantic records. Please rephrase. Also, punctuation is jumbled. * Line 30. "serves as data to validate", not really clear, could you reformulate? * Pag 15 line 19. An anthropogenic signal shouldn't be expected only for the recent decades, as humans and colonization of South Africa were active (and potentially modifying the vegetation) also during the LIA, please check/reformulate. * Conclusions: please reconsider the use of resounding wording like "unique" (twice; surely this is not the only record to report SSTs along with terrestrial proxies), "not only" * Caption Fig. 4. Explain what the 10,000 iterations are. Also, please turn the numbers of the y-axis by 180 degrees. * Fig 5. Why no LIA grey block until the right part of the figure? It seem that the last curve to the right extends into the future. MCA and LIA colour references in the caption are wrong. Also, you plot PCA but refer in the text to PC1. In general, check the wording and concordance between caption, figure and what reported in the main text, as there are several mismatches. Since there are many proxies, you should avoid confusing the

reader with slightly different wordings. * Fig. S2 contains mistakes: commas instead of points, no units for temperatures, BIT-index instead of BIT.

---

## Author Comment (AC1) · 30 Jan 2017

**Rebuttal Anonymous Referee #1**

Thank you to the reviewer for the helpful comments on this manuscript! We have tried to include the helpful comments of the reviewer in the modified manuscript (for the moment only the modified figures could be attached). This study represent a multi-proxy approach based on a marine sediment core GeoB18308-1, located on the South Coast of South Africa, offshore the Gouritz River, Southern Cape. The study reconstructs approximately the last 4 ka and additionally presents samples inland within the catchment areas of the Gouritz River itself. The authors interpret their data as demonstrating humid conditions in the Gouritz River catchment during the Medieval Climate Anomaly with lower, but highly variable sea surface temperatures in the Mossel Bay area. On

the contrary they claim that the Little Ice Age was characterized by relatively warm sea surface temperatures in Mossel Bay and arid climatic conditions favorable to torrential flood events sourced in the Gouritz headlands. I am generally excited about this work as it shows new data from an area missing detailed marine/terrestrial records. I think it is a very detailed and solid approach particularly having material from the Gouritz River catchment for "ground-truthing" in a source to sink approach. I am also okay with the conceptional model explaining the atmospheric circulation system which is based on A.L. Cohen and P.D. Tyson 1995. I am in favour of publication of this record, however would like the authors to respond/check some aspects of the paper which I describe below. My main two concerns are the construction of the age model and the interpretation of the data in the LIA. Age Model First of all, if two labs are used, Poznan and Beta lab in this case, it should be shown that the results are consistent between labs. Has a comparison on an aliquot sample been done which shows that both labs come to the same conclusion? Unfortunately, no comparative study of the 2 C14 labs has been done and we have no remaining funds to do so. However, both labs have assured us that they follow stringent procedures to ensure the quality and reproducibility of their results. Table 1 shows all material dated but only Figure 4 caption reveals what was used for the age model thereafter. It should be clarified in table 1, which of the core depths were not part of the age model. As I understand depths 123 cm, 285 cm and the reworked package at 26-66 cm depth were excluded from the age model. we have added this information also to Table 1 Hence the levels taken into account are 16.5 cm. 69 cm, 125 cm etc. Core depth 69 cm, TOC measured, gives a calibrated age (median) according to table 1 of 1294 cal. Age BP. The level used thereafter is 125 cm, TOC measured, gives a median of 598 cal. Age BP. Also the two levels below are significantly younger than core depth 69 cm dated. Why were these samples part of the age model and not excluded although they could be equally reworked material? Sorry there is a mistake in the caption of Figure 4: the reworked package is at 26-69 cm depth and (not 66cm)! The 1294 cal. Age BP age at core depth 69 cm, was removed from the age model as part of the redeposited package. In fact the TOC sample at 125 cm just plots
outside the uncertainty level given by the Baysian age model. Normally this software should give a probability estimate stating how likely this date is part of the age model or not. I somehow also see a mismatch between the table 1 data and Fig. 4. For example the plot shows two TOC point just below the blue shaded area 'reworked package, erosional contact 0. I believe that this refers to the sample at 60 and 69 cm core depth according to table 1. However, the author writes that samples between 26-66 cm were excluded. So the sample at 60 cm should not be in there. Moreover, 490 cm core depth has a median age of 4720 cal. Age yr BP which is not even part of the axis in Fig. 4. And there are more examples C2 were the cal. age from table 1 does not fit the cal. age on the axis of Fig. 4. If the author could clarify this mismatch and the core depths used and revise. We thank the reviewer for pointing out these errors! There were errors in Tab 1 and Fig. 4 that were responsible for this mismatch. They have been corrected. It is not clear to me why one would calibrate with an SHcal and then with the marine 13 in the core intervals below despite high BIT index in that interval? Moreover, the BIT index wasn 0 t even measured on the same samples the TOC was dated. We understand that this is an unusual approach, it is unfortunately a consequence of working at the marine-terrestrial interface since the TOC in such a nearshore depositional area is bound to be a mix of terrestrial and marine material. This we had to take into account when calibrating our C14 dates. However, we have no way of determining the exact percentage of marine our terrestrial material in each dated TOC sample - our choice of a calibration curve has to be based on interpretations of the available data. Compound specific dating was unfortunately also outside our budget. The solution we offer is therefore relying on the (what we think) best interpretation of the available data: using our various indirect parameters (XRF data, sediment color) and a direct indicator of soil input (the BIT index) we have identified sediment intervals that are marine and other that are fluvial deposits and chosen the calibration curves accordingly. We believe that resulting age model is reliable enough for the scope of this paper, but we are open to suggestions that will help us improve our age-depth estimations! Why was there no radiocarbon dating conducted on foraminifera from the same material? Unfortunately,
dating foraminifera ages are not available here, this has financial reasons, but also because plantic foraminifer do not live at these shallow depths and benthic species are prone to recording the 14C signal of old bottom water masses. Interpretation of the LIA interval: I am not sure the data during LIA supports the claim made for that interval. The author states that there is missing age control in that period due to re-depositional events. So no data is shown. Instead the author concludes that based on redeposited material which can be characterized as reworked soil, the time frame of the LIA must have had torrential rains and flashfloods on the background of an arid climate. I can 0 t see the evidence for that conclusion We agree with the review that we draw this conclusion very swiftly – it draws on ideas from the catchment sample analysis 5.1.1. that we neglected to refer to in the discussion of the LIA climate (5.2.3). We hope that the revised version of 5.2.3. (below) presents the evidence for an LIA Arid climate with flashfloods more clear: Continuous sedimentation at the core site was interrupted at  $\sim$ 650 cal vr BP. One or more erosive event deposits are inferred from the sedimentology (erosive contact at  $\sim$ 60-65 cm; fine sand with intercalated organic layers and lumps from  $\sim$ 30-60 cm) and an age reversal indicated by the 2 radiocarbon dates in this interval ( $\sim$ 1,466 cal yr BP at a depth of 31 cm  $\sim$ 640 cal yr BP at a depth of 60 cm and; Fig. 4; Table 1). Due to the discontinuous nature of the deposition we are only able to curtail the timeframe of deposition to having taken place between  $\sim$ 650 cal yr BP (the youngest age in the event deposit) and a post – bomb date  $\sim$ 13.5cm above the redeposited. From the redeposited sediment package 3 samples have been analyzed organic geochemistry (see Table 3). The average values over the possible timeframe of deposition is plotted in Fig.5. The high BIT-index ( $\sim$ 0.7) indicates that the redeposited package can be characterized as reworked soil material. The averaged  $\delta$ DC31 signature of ~-135‰ is comparable to that of Gouritz river paleoflood deposits described in section 5.1.1. We therefore suggest that the origin of the event-deposited material in core GeoB18308-1 is similar to the origin of these terrestrial paleoflood deposits. Our catchment study (Leaf wax  $\delta$ DC31 of paleoflood and soil deposits-see 5.1.1) indicates that paleoflood deposits are primarily induced by an increase in high
latitude precipitation i.e. precipitation in the upper parts of the Gouritz catchment. The shift in  $\delta$ 13CC31 towards slightly more depleted values in the event deposited material in core GeoB18308-1 (average in the redeposited unit: ~-28‰) furthermore indicates that the n-C31 alkanes contained in the event deposit were produced by plants under less water stress (c.f. Collister et al., 1994, Ehleringer and Cooper, 1988) than those deposited before ~650cal yr BP. We therefore infer an increase in upper catchment rainfall inducing floods for the time period of the event deposit(s) ( $\sim$ 300-650 cal yr BP). This roughly falls into the timeframe of the so-called "Little Ice Age (LIA)" recorded as humid throughout the South African WRZ (Meadows et al., 1996; Benito et al., 2011; Stager et al., 2011; Weldeab et al., 2013) due to a northward shift of the SHW (Tyson and Preston-Whyte, 2000; Chase and Meadows, 2007). In the uppermost Gouritz catchment (Seweweekspoort site) a major SHW sourced rainfall regime has been documented (Chase et al. 2015). Desmet and Cowling (1999) indicate that despite the general SRZ regime in the Gouritz catchment, the SHW supply additional rainfall in extreme events. We suggest that an increase of these extreme SHW-sourced rainfall events produced large floods during the LIA (~300-650 cal yr BP)....

neither for the claim, that the SST 0 s in Mossel Bay were warm if there is no TEX data for that core depth presented. The reviewer is correct; we were not able to calculate SSTs for the LIA timeframe due to the confounding influence of the very high soil content in this interval. The warmer SSTs in the LIA is merly something we suggest as a consequence of applying the Cohen and Tyson model to our findings. We see how this is misleading and have reformulated the abstract accordingly and we have removed the following sentence from the conclusion: In contrast; a weakened, more northerly SIA (e.g. during LIA conditions) has the opposite effect: the weaker Agulhas current is less liable for upwelling and the more frequent SHW advect warm surface water plumes onto the Agulhas bank in analogy to the modern day winter situation Moreover, what does the average line for deposit mean for the interval shown in Fig. 5? The average line for deposit represents the averages of all the measurements made in the redeposited sediments. To make this clearer we have modified Fig. 5 and the caption
accordingly. The presentation of the LIA climatic / oceanographic data is difficult due to the lack of age-control in this part of the core. We have however collected data from the redeposited interval that we believe represents the LIA climatic / oceanographic conditions. They can however not be plotted against time in figure 5 since these are not continuous, but event deposits. WE therefore opted for presenting averages of the measured data points in the event deposits. Obviously we have done a bad job in presenting this data. This leads the reviewer to enquire what the average line for deposit means. We hope the modified fig. 5 is easier to read. The SST conclusion in this paper is with odds of Zinke et al., 2014 (Zinke, J., B. R. Loveday, C. J. C. Reason, W. C. Dullo, and D. Kroon (2014), Madagascar corals track sea surface temperature variability in the Agulhas Current core region over the past 334 years, Sci. Rep., 4. doi: 10.1038/srep04393) who show that Agulhas Current SSTs cooled through the Little Ice Age. How can these opposing findings be explained? Moreover, I think there should be more evidence for that claim presented. Thank you for this reference. As stated above we have no SST data for the LIA timeframe so unfortunately we have no basis for a discussion. 2) Specific questions/issues: Page 2 line 4: I feel that a more African specific chapter of the IPCC report should be cited here rather than Metz or Kirtman et al : "Niang, I., O.C. Ruppel, M.A. Abdrabo, A. Essel, C. Lennard, J. Padgham, and P. C3 Urguhart, 2014: Africa. In: Climate Change 2014: Impacts, Adaptation, and Vul- nerability. Part B: Regional Aspects. Contribution of Working Group II to the Fifth Assessment Report of the Intergovernmental Panel on Climate Change [Barros, V.R., C.B. Field, D.J. Dokken, M.D. Mastrandrea, K.J. Mach, T.E. Bilir, M. Chatterjee, K.L. Ebi, Y.O. Estrada, R.C. Genova, B. Girma, E.S. Kissel, A.N. Levv. S. MacCracken, P.R. Mastrandrea, and L.L. White (eds.)]. Cambridge University Press, Cambridge, United Kingdom and New York, NY, USA, pp. 1199-1265. Done Page 2 line 20: There are more recent studies by now showing insolation driven responds of Southern Africa climate and should be cited here: (Daniau, A.-L., M. F. Sánchez Goñi, P. Martinez, D. H. Urrego, V. Bout-Roumazeilles, S. Desprat, and J. R. Marlon (2013), Orbital-scale climate forcing of grassland burning in southern
Africa, Proceedings of the National Academy of Sciences, 110(13), 5069-5073. doi: 10.1073/pnas.1214292110); (Simon, M. H., M. Ziegler, J. Bosmans, S. Barker, C. J. C. Reason, and I. R. Hall (2015), Eastern South African hydroclimate over the past 270,000 years, Scientific Reports, 5, 18153. doi: 10.1038/srep18153) We have added Daniau et al. 2013 and Simon et al. 2015 Page 3 Line 1: Biastoch et al., 2009a does not show that strong SHW reduce leakage into the SA and should not be cited here in this respect. This study only shows what effect shifting the SHW to Leakage strength has. It does not evaluate what a change in the strength of the SHW does to leakage variability. In this respect the citation of Durgadoo et al., 2013 in the line below is wrong as in this paper the authors show that an equatorward shift in westerlies increases leakage and not like written in this paper page 3 line 4:" a weakening of the Agulhas Current and the leakage of warm water due to northward displacement of the SHW". Both references were removed Page 5 line 21: Not sure how this description of the bathymetry fits into this part of the oceanography. Would suggest shifting that. The descriptions of the bathymetry have been shifted to be included in the section 3.1. Page 6 line 29: Is that a valid common method to calibrate XFR scans? I would rather think that taking sub-samples and analyzing for bulk major and trace elements would be the way to do it? One approach could be following the below: "Prediction of Geochemi- cal Composition from XRF Core Scanner Data: A New Multivariate Approach Including Automatic Selection of Calibration Samples and Quantification of Uncertainties By G. J. Weltje, M. R. Bloemsma, R. Tjallingii, D. Heslop, U. Röhl and Ian W. Croudace." Yes we did take sub-samples and analyzed them for bulk major and trace elements, it is not expressed clearly in our methods section, but we have added that 28 dried and ground subsamples were analyzed for bulk major and trace elements for calibration purposes. Page 8 line 19: Why and how was the original method modified? Does the modification have advantages compared to Hopmans protocol? If so that should be stated there. This can be answered in the methods section (expert co-authours: Gesine Mollenhauer and Enno Schefuss). Page 9 line 27: To be statically significant one have to at least count 150 specimens per sample

**CPD**
not only 20. This can be answered in the methods section (expert co-authours: Peter Frenzel and Stephanie Meschner). Page 12: By which evidence sedimentological etc. was a paleosoil and a flood de- posit distinguished? Only by different dD values? That should be better described and presented in the text. Soil horizons and flood deposits were also distinguished by their sedimentology in the field. We have added the following information to 5.1.1.:"Catchment samples were all taken at lowland locations, however some were identified as soil samples from horizons of darker, finer material while others were identified as flood deposits by their lighter, coarser facies (Fig. 6)." Page 12 line 10: why would rainfall in the highlands auto- matically lead to flood events? We do not infer that this is automatically so in every catchment, but it seems to be the case in the Gouritz catchment; the layers that we have identified sedimentologically as flood deposits contain organic material that was synthesized under different conditions than the plant material contained in the soils. The deuterium values of the flood deposits are depleted – this gives an indication of their origin as rainfall becomes deuterium depleted with origin. We hope to have made this chain of thought clearer in the text. Page 14 line 15: If that is stated then values should be given as well. As the age model was derived from a Baysian approach one can give an uncertainty value here for the age model. Added: (+/-2Æą: 835-1100cal yr BP) Page 14 line 17: The recent review paper by Should be included in that part of the manuscript. Also Nash, D. J., G. De Cort, B. M. Chase, D. Verschuren, S. E. Nicholson, T. M. Shanahan, A. Asrat, A.-M. Lézine, and S. W. Grab (2016), African hydroclimatic variability during the last 2000 years, Quaternary Science Reviews, 154, 1-22. doi: http://dx.doi.org/10.1016/j.guascirev.2016.10.012 Woodborne, S., G. Hall, I. Robertson, A. Patrut, M. Rouault, N. J. Loader, and M. Hofmeyr (2015), A 1000-Year Carbon Isotope Rainfall Proxy Record from South African Baobab Trees (Adansonia digitata L.), PLoS ONE, 10(5), e0124202. doi: 10.1371/journal.pone.0124202 should be added here as their record also shows the wettest period was c. AD 1075 in the Medieval Warm Period. Nash et al., 2016; Woodborne et al., 2015 included Fig. 5: For comparative purposes other regional paleoenvironmental records are plot-
ted. How was secured that there are no age model offsets between this study and the other records? We see the concern of the reviewer – no age model is prefect, but each record is based on a relatively reliable (published) independent age depth model. We do not see how we can improve this. 3) Technical corrections Fig. 2: page 29 line 5: legend says Carr et al., 2014 in the figures in the map it says Carr et al, 2015 Done Fig. 5 Could do with more labels on the Y-Axis i.e. at least 500 year tick labels between tick marks. Done Page 12 line 12: twice 'mainly used here! Rephrase grammar is wrong in that part of the sentence! Done Page 12 line 26: Formatting issues and missing space. Done Page 13 line 21. Fig. 5 shows the main record only till 4 ka according to the axis however the text states: " The oldest part of the 18308-1 paleorecord ( âĹij 4880-1150 cal yr BP) Has been corrected to 4058 cal yr BP ...... where is the rest of the bottom of the paper And what are the dots in SF1? The calibration samples or the subsampling for the organic geochemistry? No, the discrete measurements (in mg/kg) for calibrating the XRF scans are plotted as squares. Č

Please also note the supplement to this comment: http://www.clim-past-discuss.net/cp-2016-100/cp-2016-100-AC1-supplement.pdf

**Supplement:**

[Figure]

**Fig. S1 XRF-data of the GeoB18308-1 plotted against depth (m). Scanning data (cts) is indicated by the curve whereas discrete measurements (in mg/kg) are plotted as squares.**

[Figure]

Fig. S2 BIT and SST (inferred from TEX$_{86}$$^{H}$) bipolt showing a lack of correlation and thus excluding the possibility of high terrestrial input.

[Figure]

**Fig. S3** Crossplots for axes 1 and 2 of a Principal Component Analysis (PCA) on microfossil distribution in core GeoB18308-1. High positive loadings of freshwater ostracods on axis 1 explaining 35.1% of variation. We assume input of continental material to be the best descriptor of this component. Axis 2 explains 22.4% of variation and is characterized by suspension feeders as the foraminifer Lobatula lobatula and sand dwelling species as the ostracod Neocytherideis cf. boomeri at high loadings or planktic foraminifera sedimenting in calm water at low loadings. The probable explaining environmental factor for this axis is turbulence of the water or input of particulate organic matter.

---

## Author Comment (AC2) · 30 Jan 2017

The study encompasses an impressive variety of methodologies and of different proxies, and discusses the rich results in a convincing way, especially for what regards the aridity-humidity multi-proxy reconstruction. An undoubtable strength of the approach is the analysis carried out in samples from several places in the Gouritz catchment, which provides decisive supports for the inferences made. I would recommend that the manuscript be published, although prior to that the authors should improve a few aspects of it, and respond to some questions that I detail below. Thank you to the reviewer for the helpful comments on this manuscript! We have tried to include the helpful comments of the reviewer in the modified manuscript (for the moment only the modified figures could be attached). General reservations that I have with this work are: 1) a better effort could be made of emphasizing, especially in the abstract

and introduction (and potentially also in the title), what the key findings are and what their importance implications of their results is. We have added the implications of the key findings at the end of the abstract. We have modified the introduction as well, but we thought it best fit to stress the implications of the key findings mainly in the abstract. As it is it resembles more an account of analyses carried out in a very good setting (whose importance could be made even more clear). We have added the following to the beginning of the abstract: "In addition to this, it's location at the interface of Atlantic and Indian Ocean circulation systems makes the southernmost tip of South Africa a climatically extremely sensitive as well as interesting area. Thus far few marine records have been available in order to study the interplay of marine and atmospheric circulation systems. This study of sediment core GeoB18308-1 at the terrestrial-marine interface fills this gap for the time interval of ~4 ka BP." 2) It would be very interesting if the authors could draw more explicitly the implications of their results, and/or of the conceptual model they somehow validate. To explain better what we mean by validate we have added: "The only SST record published for the area (Cohen and Tyson, 1995) does not include data for the time frame in question."....." The decrease in SSTs recorded in GeoB18308-1 for the interval of increased humidity in the Gouritz River catchment inferred in this study for the time interval of the Medieval Climate Anomaly serves as data to validate the conceptual model by Cohen and Tyson, (1995) for which thus far no Medieval Climate Anomaly data had been available." for the latitudinal shifts in ITCZ. This is of interest to a larger climatological community, and to projections of what the region may expect with ongoing climate change. We did not want to include an attempt to make a predictions of future climate that are too speculative. We have however added a suggestion of a future prognosis at the end of the abstract to address this reviewer comment. 3) the paper is very wordy, especially in its sections 2 to 5. The authors should improve readability and really consider refraining from reporting all they have done and all results, and focus of what is of relevance to the new findings discussed. Some records barely matter for the discussion. We have tried to shorten the complete regional settings section (2) as
well as the methods (3) and results (4) to a minimum in particular for the not much used heavy mineral and microfossil proxies. Furthermore we have rewritten large parts of section 5 in order to make it more readable (even if maybe not shorter) 4) Probably because of the vast amount of material presented, the manuscript is sloppy in many parts: odd sentences, mismatches in the wording, punctuation, typos. One would expect that nine authors could proofread the manuscript to a higher quality. We have reworked the manuscript to fix this. Main specific points The first sentence of the introduction seems inconsequential and unjustified to me: there is no argument for the importance of South Africa's geographic position. Re- phrase. We have rephrased the first sentence to make this clearer. I would suggest to pay more attention to streamlining the introduction chap- ter: as it is it is hard to read, and the main points that the authors wish to make do not come through clearly. What are the main research gaps regarding South African climate? Can you present the evidence for one or the other explanation in a more or-ganized manner? We have removed some of the surplus information and focused the introduction on the 2 main research questions of the region. I would recommend an effort to focus section 2. It could be made more concise, and thus the readability of the paper could improve, if you privilege the information that is relevant to the findings of this paper. E.g., the reader doesn't gain insight that are relevant to this Late Holocene paper by your discussing Cretaceous tectonics. We have removed some of the surplus information, the section now only includes what is relevant for the interpretation of the data. Fig 4. The LIA follows the MCA, not the other way around. Changed Pag 14 line 2 and following. First, from fig 5 one would say that all discussed changes happen from ca. 950 yBP, rather than 1150. Can you clarify whether the figure or the discussion are correct? The figure is correct I have changed the text Further, I don't think you can state that anything happens to the SST record around 1150 yBP, at least from the results contained in fig.5. Simply the sampling temporal resolution increases, but I would argue there is no real difference in variability before and after 1150 kBP. If anything, low peaks appear after that time: could you show a real statistical significance between the average SSTs either side of
1150kBP? None of the SSTs inferred for the period before the MCA are lower than the average SSTs measured for the MCA interval (n=17). However, we will remove all mention of a difference in variability, this may indeed be an artefact of the sampling resolution as the reviewer suggest... (related to one of the main objections of reviewer 1) You state that age-reversals oc- cur from 650 yBP, but from figure 5 one can see that you take data up to ca. 500 yBP seriously (also the gray band starts at 500): can you clarify? Sorry, there is a mistake in the caption of Figure 4: the reworked package is at 26-69 cm depth and (not 66cm)! The 1294 cal. Age BP age at core depth 69 cm, was removed from the age model as part of the redeposited package. Pag 14 II 27-28. For intensification of Agulhas Current transport, you should check Durgadoo et al (2013), who report, from three ocean models, that northward shifts (and intensification) in at- mospheric features increase Agulhas Current transport contrary to what included in the conceptual model here discussed. This does not mean that ocean models in the above studies hold the truth, but I would suggest you could take the occasion to discuss this contradiction in the literature. (also in the Conclusions) Pag 16 line 10. Future climate change may follow what pattern? You reported two. We have removed the LIA scenario so it is hopefully clrear now that we are referring to the MCA scenario. Also, could you provide a reference supporting this? (rephrase anyway, as sentence is confused) Unfortunately not, this is our suggestion... Minor comments: The title could be modified to eliminate the present continuous tense - vague - and include any word that reports the results of this "linking" New title: Southerly anticyclonic circulation drives climatic conditions and sea surface temperatures in southernmost Africa Abstract: "highly dynamic" and "highly complex" used just one sentence apart, maybe either make more specific or eliminated one Done "give information on climatic changes": it is vague, make more concrete. Oceanographic and hydrologic changes in specific The last sentence is unclear: to which processes do you refer, to those in the LIA or in the MCA? Rephrase. Also, probably not appropriate to only refer to a climate model like this at the end of the abstract, where the reader cannot make much out of it: essential information is missing. We have changed the abstract accordingly
LI 9-12. This sentence is complicated and doesn't show a contrast between concepts that one would expect from the use of "while". LI 16 ITCZ not explained, maybe avoid abbreviation as never used anymore. Written out LI 16-18 you either use whether, or add a question mark, not both. Question mark removed Page 3 line 4. Durgadoo et al 2013 find precisely the opposite, i.e. that Agulhas leakage increases when westerlies move north. I would suggest you deal with this in the introduction. As also suggested by referee1 the citation was removed line 6. What is YRZ? Written out in text LI 16-17. Odd phrasing, a sediment core doesn't aim to anything. Rephrased to "our work" aims Line 28. Harmonize the units (use exponential in place of Mm3) Done Line 32. What do you mean by mixed summer and winter rainfall? And why this single paleo piece of information in a present-day context? Removed Page 6 II 15-16. Sentence not clear: what is the unit for the numbers in parenthesis, years? (14C should have 14 in the superscript) Dewar et al 2012 is missing from the ref list. Unit and reference added Line 17. Why do you inform about the sedimentation rate: this is not further discussed in the paper. Removed Line 21. Analyses. Line 24. Change "elemental profiles" in place of "scanning data". Done Line 27. Change "vertical resolution or downcore resolution" in place of depth resolution. Done What are 1.2 cm2? Removed Line 3. Change "scanning intensities" for a more appropriate term Changed to peak intergals Pag 7 II 4-5. Not clear how the xrf data helped in selecting the samples for organic geochemistry, please reformulate. Reformulated – a higher resolution is chosen in the upper part.... Methods: try to avoid so many abbreviations, especially those not further used in the paper. In general the manuscript is highly packed with abbreviations, please try to be parsimonious with them. We tried to stil to this advice throughout the manuscript Pag 9 line 16. "micropaleontologically" probably not a word. Changed to microfossil analysis Pag 10 line 2. "None of the considered taxa was found to correlate". Added Line 4. What do you mean by point counted? Point counted is a method used instead of grain counting of minerals in order to do justice to the larger size of some minerals - Because some researchers are probably not truly informed about this fundamental technique, we have changed the text... As you can see just

**CPD**
few rows after it is explained the importance of this method. ... I can also suggest to have a look at this nice paper of a colleague of mine for a wider application to your specific studies. Garzanti Eduardo 2016. From static to dynamic provenance analysis—Sedimentary petrology upgraded. Sedimentary Geology 336, 3–13. He wrotes at page 5/11: "Point-counting techniques are highly recommended for heavy-mineral analyses carried out on bulk-samples or wide grain-size windows. This is because the discrepancy between the real volume percentages and the number percentages as determined by grain counting increases with the increasing width of the grain-size window analyzed, and volume percentages are systematically overestimated for denser minerals that are smaller than settling-equivalent lower-density minerals (Galehouse, 1971)." This reference is alreday included in the manuscript: Galehouse, J.S., 1971. Point counting. In: Carver, R.E. (Ed.), Procedures in sedimentary petrology. Wiley, New York, pp. 385–407. Line 15. Same reference occurs twice. Removed Line 20. Fig. S4 does not exist. S3 Pag 14 II 15-17. The reader gets the impression that the MCA is a Southern African phenomenon, while this is a concept normally applied north Atlantic records. Please rephrase. Also, punctuation is jumbled. We have clarified that this is a NH trend expressed in South Africa.... Line 30. "serves as data to validate", not really clear, could you reformulate? Rephrased: The decrease in SSTs recorded in GeoB18308-1 for the interval of increased humidity in the Gouritz River catchment inferred in this study for the time interval of the Medieval Climate Anomaly serves as data to validate the conceptual model by Cohen and Tyson, (1995) for which thus far no Medieval Climate Anomaly data had been available. Pag 15 line 19. An anthropogenic signal shouldn't be expected only for the recent decades, as humans and colonization of South Africa were active (and potentially modifying the vegetation) also during the LIA, please check/reformulate. We see no evidence of anthropogenic impact so I have removed this entirely Conclusions: please reconsider the use of resounding wording like "unique" (twice; surely this is not the only record to report SSTs along with terrestrial proxies), "not only" removed / changed to "advantage" Caption Fig. 4. Explain what the 10,000 iterations are. Caption modified: the calibrated 14C dates

**CPD**
(transparent blue) and the age-depth model (darker greys indicate more likely calendar ages; grey stippled lines show 95% confidence intervals; red curve shows single 'best' model based on the weighted mean age for each depth). Also, please turn the numbers of the y-axis by 180 degrees Done Fig 5. Why no LIA grey block until the right part of the figure? It seem that the last curve to the right extends into the future. MCA and LIA colour references in the caption are wrong. Also, you plot PCA but refer in the text to PC1. In general, check the wording and concordance between caption, figure and what reported in the main text, as there are several mismatches. Since there are many proxies, you should avoid confusing the reader with slightly different wordings. We have tried to pay attention to this changing Fig. 4 and 5 Fig. S2 contains mistakes: commas instead of points, no units for temperatures, BIT-index instead of BIT. We have modified the Fig S2 accordingly.

Please also note the supplement to this comment:

http://www.clim-past-discuss.net/cp-2016-100/cp-2016-100-AC2-supplement.pdf
Fig. 1.
Fig. 2.

**Supplement:**

[Figure]

**Fig. S1 XRF-data of the GeoB18308-1 plotted against depth (m). Scanning data (cts) is indicated by the curve whereas discrete measurements (in mg/kg) are plotted as squares.**

[Figure]

Fig. S2 BIT and SST (inferred from TEX$_{86}$$^{H}$) bipolt showing a lack of correlation and thus excluding the possibility of high terrestrial input.

[Figure]

**Fig. S3** Crossplots for axes 1 and 2 of a Principal Component Analysis (PCA) on microfossil distribution in core GeoB18308-1. High positive loadings of freshwater ostracods on axis 1 explaining 35.1% of variation. We assume input of continental material to be the best descriptor of this component. Axis 2 explains 22.4% of variation and is characterized by suspension feeders as the foraminifer Lobatula lobatula and sand dwelling species as the ostracod Neocytherideis cf. boomeri at high loadings or planktic foraminifera sedimenting in calm water at low loadings. The probable explaining environmental factor for this axis is turbulence of the water or input of particulate organic matter.

---

## Referee Report (RR1)

Review of "Southern hemisphere anticyclonic circulation drives climatic conditions and sea surface temperatures in southernmost Africa" (re-submitted) by Hahn et al.

I am generally satisfied by the authors' addressing of both my main and my minor suggestions.

I have one pending issue related to the scientific content of the paper: in the conclusions the authors include a short sentence about the possibility of paleo scenarios replicating due to climate change. I think this is not by any means well articulated in the paper, and should be discussed to some extent, either in the (end of the) discussion, or in the conclusions. It is critical for the reader to know that it is actually suggested: precisely what scenario may replicate, and why should we think that this is plausible, given what is known about present and future climate change (ITCZ displacement, circulation patters changing,…).

Last, I suggest the authors should pay close attention in revising the writing, as this seems to still require attention. I will use the most outstanding parts of the article to exemplify the type of issues that should be checked:

- Title: I leave this ultimately to the authors and the editor, but isn't it misleading to use the present tense to refer to past phenomena? I would use "drove Holocene climate" (so also the timeframe is specified) instead of "drive".
- Abstract: I don't see a real "contrast" between "climatic conditions favorable to torrential flood events" on the one hand, and "lower sea surface temperatures" and "humid conditions" on the other.
- Abstract line 18: I don't see the need to use the resounding word "unique".
- Abstract line 28: wrong verb conjugation. In general verb tenses are switched seemingly randomly throughout the abstract, i.e., events of the past are referred to in either present or past tense (e.g. lines 23-30). Please correct this (also check the rest of the manuscript).
- Conclusion line 34: not sure whether "The down-core GeoB18308-1" can be considered the subject of a sentence.
- Conclusion line 36: it seems like the LIA continues until present.
- Conclusion line 38-39: this seems a repetition of lines 28-29
- Conclusion line 1 (second page): it doesn't seem useful to introduce acronyms in the conclusions.

---

## Author Response (AR2)

Reviewer comments in black - Rebuttal in blue

**REVIEWER 1**:

Suggestions for revision or reasons for rejection (will be published if the paper is accepted for final publication)

Review on revision of "Southern hemisphere anticyclonic circulation drives climatic conditions and sea surface temperatures in southernmost Africa" by Hahn et al. Thanks for commenting on my remarks and questions. Many things have been clarified. I still have some concerns to raise: We again thank the reviewer for his/her time and very helpful ideas.

• Figure capture 4 still has the wrong depth for the "the reworked package at 26 – 66 cm depth" in there. CHANGED

• As the paper does not have SST data for the LIA timeframe I would be careful with stating in the title of the paper that "Southern hemisphere anticyclonic circulation drives sea surface temperatures in southernmost Africa." Moreover, with the current title it is not clear what time interval this study covers. Clearly, this is not a modern day study. CHANGED

 In general, what would shift the SHW during the LIA northward? That could be interesting point to add for the reader. Especially, as a recent compilation (Lechleitner et al., 2017 : "Tropical rainfall over the last two millennia: evidence for a low latitude hydrologic seesaw." DOI: 10.1038/srep45809) has shown that during the LIA the ITCZ was generally shifted further south causing wetter conditions in the SH. A more southerly ITCZ would also prompt a southward shift of the subtropical anticyclones hence causing the opposite effect in South Africa during the LIA as stated in the conceptual model explained here. How can these two views be reconciled? Thank you for indicating this new study by Lechleitner et al.! We are happy to discuss this in the LIA sections 5.2.3 2 "Conditions after ~650 cal yrs BP – Little Ice Age and beyond". "We therefore infer an increase in upper catchment rainfall inducing floods for the time period of the event deposit(s) ( $\sim$ 300 – 650 cal yrs BP). This roughly falls into the timeframe of the socalled "Little Ice Age (LIA)" recorded as humid phase throughout the South African Winter Rainfall Zone (Meadows et al., 1996; Benito et al., 2011; Stager et al., 2011; Weldeab et al., 2013). Most regional studies associate this increase in rainfall with a northward shift of the SHW (Tyson and Preston-Whyte, 2000; Chase and Meadows, 2007). A global study using high resolution records (Lechleitner et al. 2017) also documents a southward shift of the ITCZ during the LIA interval; however no data from the southern hemispheric mid-latitudes is included and we suggest that the effects of this shifts are limited to locations north of our study site." . It is very unfortunate that we have no SST record for the LIA timeframe in order to further test the conceptual model using the Leitner et al. data.

• I am not entirely happy with Figure 5. The way it is now, it is hard to see what indications there are for what environmental trend. Perhaps plotting age on the x-axis and the parameters as stacked y-axis would improve it. Also, giving the parameters arrows for trends will help to see better. I would plot things together which are indicating the same environmental interpretation. Changes made in Fig. 5 I don't really see how g,h,i and j panels are adding anything. Aridity in g) is not changing at all throughout the record. It is true that the shift in  $\delta^{13}C_{C31}$  is minimal, however none of the  $\delta^{13}C_{C31}$  values inferred for the period before the MCA are higher than the average value

measured for the MCA interval (n=27) and we do believe that this slight shift is worthwhile discussing: it does not indicate a vegetation change in the catchment (obvious the timescale is much too short for this) but it can indicate a decrease in plant water-use efficiency as the vegetation adapted to slightly moister conditions. The representation of this in Fig. 5 is not good; we have adjusted the axis scaling and also addressed the issue in the text (sec 5.5.2): "Heavier isotope 13C is more depleted in C3 plants which are less adapted to aridity as opposed to C4 (Collister et al., 1994). However, the observed shift is very minor and a change in vegetation type is unlikely to occur within the timescale in question. It is more probable that the  $\delta$ 13C values became more depleted as the water-use efficiency of plants decreased in moister climatic conditions (Pate, 2001; Ehleringer and Cooper, 1988)."h) temperature- if you would add trend lines through the data you could see two steps: lower T until 2500 ka BP and then slightly higher values on average to core top. I don't see how the Congo Cave data helps with the LIA and MCA interpretation of this study? Yes, we see this point; we will remove the Cango Cave record as suggested however for comparative purposes we would like to show the Chase et al. and Wündsch et al. records. Potentially just plotting the data from that study together with an axis break in the 4000-1000 ka BP range would also blow up and highlight the LIA MCA data/interval more/ make it more visible !? We have also considered using an axis break from 4-1 ka BP, however that would inhibit us from showing the high resolution grainsize and XRF data for this time interval. This data is relevant because it indicates that conditions were actually stable during the period in question which in turn "justifies" our choice of resolution for the biomarker analysis in this timeframe. Furthermore, the observation of stable conditions during the 4-1ka BP timeframe is relevant in itself and "in accordance with the relatively stable conditions recorded in the millennia prior to ~1000 cal yr BP in the speleothem layer width from the Cold Air Cave (Holmgren et al., 1999) as well as pollen data from Wonderkrater and Rietvlei Dam (Scott et al.. 2012) in southeastern Africa." I would like to encourage the author to think about a better way to show the data in Figure 5 as this is the main dataset the paper is based on. I think Figure 3 and 7 can be supplement material. We thank the reviewer for these suggetsions on improving Figure 5 and we hope that the results is more reader – friendly. We also agree that Figs.3and7 can easily be moved to the supplements.

**REVIEWER 2:**

Review of "Southern hemisphere anticyclonic circulation drives climatic conditions and sea surface temperatures in southernmost Africa" (re-submitted) by Hahn et al. I am generally satisfied by the authors' addressing of both my main and my minor suggestions. We again thank the reviewer for his/her time and very helpful ideas.

I have one pending issue related to the scientific content of the paper: in the conclusions the authors include a short sentence about the possibility of paleo scenarios replicating due to climate change. I think this is not by any means well articulated in the paper, and should be discussed to some extent, either in the (end of the) discussion, or in the conclusions. It is critical for the reader to know that it is actually suggested: precisely what scenario may replicate, and why should we think that this is plausible, given what is known about present and future climate change (ITCZ displacement, circulation patters changing,...). Already the editor has struggled with this idea – it is true that we have not included a regional climate

**model in this study – so we should not make linkages to the future – This sentence in the conclusion was going out too much on a limb. I have removed this speculative sentence.**

Last, I suggest the authors should pay close attention in revising the writing, as this seems to still require attention. I will use the most outstanding parts of the article to exemplify the type of issues that should be checked: • Title: I leave this ultimately to the authors and the editor, but isn't it misleading to use the present tense to refer to past phenomena? I would use "drove Holocene climate" (so also the timeframe is specified) instead of "drive". changed•

Abstract: I don't see a real "contrast" between "climatic conditions favorable to torrential flood events" on the one hand, and "lower sea surface temperatures" and "humid conditions" on the other. •changed

Abstract line 18: I don't see the need to use the resounding word "unique". • changed

Abstract line 28: wrong verb conjugation. In general verb tenses are switched seemingly randomly throughout the abstract, i.e., events of the past are referred to in either present or past tense (e.g. lines 23-30). Please correct this (also check the rest of the manuscript). one of the native speaking co-authours looked at this again— it should be good now...

Conclusion line 34: not sure whether "The down-core GeoB18308-1" can be considered the subject of a sentence. changed

• Conclusion line 36: it seems like the LIA continues until present. True; we have indicated that it ends ca. 300 years BP

• Conclusion line 38-39: this seems a repetition of lines 28-29 • True; line 38-39 has been removed...

Conclusion line 1 (second page): it doesn't seem useful to introduce acronyms in the conclusions. Has been removed

**Southern hemisphere anticyclonic circulation drives oceanic and climatic conditions in late Holocene southernmost Africa**

Annette Hahn1; Enno Schefuß1; Sergio Andò2, Hayley C. Cawthra3,4, Peter Frenzel5; Martin Kugel1, Stephanie Meschner5; Gesine Mollenhauer6; Matthias Zabel1

[revised manuscript text omitted]

- 15 towards lower average  $\delta^{13}C_{C_{31}}$  values in the sediment core can be observed. Heavier isotope 13C is more depleted in C3 plants which are less adapted to aridity as opposed to C4 (Collister et al., 1994). However, the observed shift is very minor and a change in vegetation type is unlikely to occur within the timescale in question. It is more probable that the  $\delta^{13}$ C values became more depleted as the water-use efficiency of plants\_decreased in moister climatic conditions (Pate, 2001; Ehleringer and Cooper, 1988). In either case,  $\delta^{13}$ C values of *n*-C31 alkanes exported from the catchment suggest a shift towards more humid
- 20 conditions on land after 950 cal yrs BP. At the same time,  $\delta D$  values of the leaf wax n-C31 alkane shift to -129‰ indicating deuterium-enriched precipitation during this time. This likely indicates a shift to lower altitude source regions. Enriched  $\delta D_{C31}$ values would be an indication for plant material mainly derived from more vegetated lowlands as opposed to plant material derived from the upper catchment during major flood events (see Sec. 5.1.1.). Within the error margin of our age-model, the shift we see at ~950 ±120 cal yrs BP towards a more humid lower Gouritz River catchment is a general eastern South African
- 25 expression of the northern hemispheric trend termed Medieval Climate Anomaly (MCA) summarized by Tyson and Lindesay (1992), Tyson and Preston-Whyte (2000) and Nash et al. (2016). A large array of continental records document this humid period throughout the South African summer rainfall zone (SRZ) (Talma and Vogel, 1992; Holmgren et al., 1999; Thomas and Shaw, 2002) as well as Wonderwerk Cave (Brook et al., 2015), Braamhoek Wetland, Free State (Norström et al., 2009, 2014), Lake Eteza, coastal KwaZulu-Natal (Neumann et al. 2010), Lake Sibaya, KwaZulu-Natal (Stager et al 2013),
- 30 Blydefontein basin, Kikvorsberge (Scott et al. 2005), Katbakkies Pass, Swartruggens Mountains, southwestern Cape (Chase et al., 2015) and northeastern South African baobab trees (Woodborne et al., 2015). The few existing records of the YRZ record similar trends; a continuous rise in precipitation was found in δ15N of hyrax middens at Seweweekspoort (Chase et al., 2013) and more evidently in the indirect (TOC-based) humidity record at Groenvlei a Wilderness lake (Wündsch et al., 2016) (Fig. 4h-i). In contrast to the large array of available continental datasets, marine records are rare. The only SST record
- 35 published for the area (Cohen and Tyson, 1995) does not include data for the time period in question. However, it does provide a conceptual model of ocean-atmospheric interplay to be tested (c.f., Fig. 6). The authors postulate a periodically strengthened and more southerly South Indian Ocean Anticyclone and South Atlantic Anticyclone having an onshore as well as an offshore effect: 1) reinforcing the tropical easterly influence causing extended warm wet spells in the South African Summer Rainfall Zone and 2) increasing the frequency of eastward ridging highs and, thus, along shore winds driving the coastal upwelling on
- 40 the eastern Agulhas Bank. Additionally, shelf-edge upwelling becomes more frequent when the strong South Indian Ocean Anticyclone increases the Agulhas volume transport. The decrease in SSTs recorded in GeoB18308-1 for the interval of increased humidity in the Gouritz River catchment inferred in this study for the time interval of the Medieval Climate Anomaly

**Deleted: Fig. 5**

**Deleted: Fig. 5**

| Deleted: towards lower      |
|-----------------------------|
| Deleted:                    |
| Deleted: (to -28.5‰) |
| Deleted: The h              |
| Deleted: Furthermore        |
| Deleted: ,                  |
| Deleted: may                |
| Deleted: o                  |
| Deleted: when a             |
| Deleted: plant's            |
| Deleted: s                  |
|                             |

| -1     | Deleted: Fig. 5 |
|--------|-----------------|
| $\neg$ | Deleted: j      |
| -1     | Deleted: Fig. 8 |

serves as data to validate the conceptual model by Cohen and Tyson, (1995) for which thus far no Medieval Climate Anomaly evidence by SST data had been available.

**5.2.3 Conditions after ~650 cal yrs BP - Little Ice Age and beyond**

Continuous sedimentation at the core site was interrupted at ~650 cal yrs BP. One or more erosive event deposits are inferred from the sedimentology (erosive contact at ~60 – 70 cm; fine sand with intercalated organic layers and lumps from ~30 – 60 cm) and an age reversal indicated by the two radiocarbon dates in this interval (~1,466 cal yrs BP at a depth of 31 cm ~640 cal yrs BP at a depth of 60 cm and; Fig. 3; Table 1). Due to the discontinuous nature of the deposition we are only able to curtail the timeframe of deposition to having taken place between ~650 cal yrs BP (the youngest age in the event deposit) and a postbomb date ~13.5 cm above the event deposit. From the redeposited sediment package three samples have been analyzed for

- 10 organic geochemistry (see Table 3). The average values over the possible timeframe of deposition is plotted in Fig.5. The high BIT (~0.7) indicates that the redeposited package can be characterized as reworked soil material. The averaged  $\delta D_{C31}$  signature of ~-135‰ is comparable to that of Gouritz River paleoflood deposits described in Sec. 5.1.1. We therefore suggest that the origin of the event-deposited material in core GeoB18308-1 is similar to the origin of these terrestrial paleoflood deposits i.e. the upper parts of the Gouritz catchment (c.f. Sec 5.1.1). The shift in  $\delta^{13}C_{C31}$  towards slightly more depleted values in the event
- 15 deposited material in core GeoB18308-1 (average in the redeposited unit: ~-28‰VPDB) furthermore indicates that the *n*-C31 alkanes contained in the event deposit were produced by plants under less water stress (c.f. Collister et al., 1994, Ehleringer and Cooper, 1988) than those deposited before ~650 cal yrs BP. We therefore infer an increase in upper catchment rainfall inducing floods for the time period of the event deposit(s) (~300 650 cal yrs BP). This roughly falls into the timeframe of the so-called "Little Ice Age (LIA)" recorded as humid phase throughout the South African Winter Rainfall Zone (Meadows)
- 20 et al., 1996; Benito et al., 2011; Stager et al., 2011; Weldeab et al., 2013). Most regional studies associate this increase in rainfall with a northward shift of the SHW (Tyson and Preston-Whyte, 2000; Chase and Meadows, 2007). A global study using high resolution records (Lechleitner et al. 2017) also documents a southward shift of the ITCZ during the LIA interval; however no data from the southern hemispheric mid-latitudes is included and we suggest that the effects of this shifts are limited to locations north of our study site. 
[revised manuscript text omitted]
 co | omposition and | distribution | descriptive | parameters | averaged | (with st | tandard | deviation). |  |
|---------|------------|-------------|----------------|--------------|-------------|------------|----------|----------|---------|-------------|--|
|         |            |             |                |              |             |            |          |          |         |             |  |

| Gample   | latitude  | longitude | δ 13 C C31 | standard  | δD c31         | standard      | Norm31 | sample material    |
|----------|-----------|-----------|-----------------------------------------|-----------|-----------------------|---------------|--------|--------------------|
| namo     | ( ] ]     | J  |                                         | deviation | 0. 17 0 1/01/1 | deviation     |        |                    |
| name     | (decimal  | degrees)  | δ 13 C C31        |           | SMOW                  | δD c31 |        |                    |
| ECT-1-2  | -34,28081 | 21,82842  | -26,77                                  | 0,05      | -127,29               | 2,69          | 0,85   | Soil horizon       |
| ECT-1-3  | -34,28081 | 21,82842  | -29,10                                  | 0,10      | -136,96               | 0,43          | 0,83   | paleoflood deposit |
| ECT-2-1  | -34,18470 | 21,75288  | -28,84                                  | 0,05      | -127,30               | 0,22          | 0,76   | paleoflood deposit |
| ECT-3-1a | -34,08183 | 21,74178  | -27,92                                  | 0,21      | -129,86               | 0,92          | 0,78   | soil horizon       |
| ECT-3-1b | -34,08183 | 21,74178  | -27,74                                  | 0,07      | -139,18               | 0,83          | 0,73   | paleoflood deposit |
| ECT-5-1  | -33,90890 | 21,65330  | -29,50                                  | 0,08      | -130,13               | 0,13          | 0,78   | soil horizon       |
| ECT-5-2  | -33,90890 | 21,65330  | -29,44                                  | 0,15      | -132,31               | 0,23          | 0,81   | paleoflood deposit |
| ECT-6-1  | -33,90917 | 21,65353  | -29,14                                  | 0,05      | -134,38               | 1,09          | 0,83   | paleoflood deposit |
| ECT-6-2  | -33,90917 | 21,65353  | -28,14                                  | 0,10      | -128,87               | 0,09          | 0,82   | soil horizon       |
| ECT-7-2  | -33,75837 | 21,46945  | -29,66                                  | 0,08      | -131,94               | 1,35          | 0,74   | riverbank deposit  |
| ECT-8-1  | -33,62422 | 22,22843  | -31,56                                  | 0,06      | -128,20               | 0,23          | 0,77   | riverbank deposit  |

|       |                      | standard                                                                                                                      |                   | standard          |        |       |                                  | SST(°C)   |
|-------|----------------------|-------------------------------------------------------------------------------------------------------------------------------|-------------------|-------------------|--------|-------|----------------------------------|-----------|
| Deptn | $\delta^{13}C_{C31}$ | deviation                                                                                                                     | δD C31 | deviation         | Norm31 | BIT   | $\mathrm{TEX}_{86^{\mathrm{H}}}$ | im et al. |
| (Cm)  | % VPDB               | $\delta^{{\scriptscriptstyle 1}{\scriptscriptstyle 3}}C_{{\scriptscriptstyle C}{\scriptscriptstyle 3}{\scriptscriptstyle 1}}$ | % VSMOW           | δD C31 |        |       |                                  | 2010)     |
| 2,5   | -28,490              | 0,284                                                                                                                         | -134,312          | 1,978             | 0,802  | 0,211 | 0,477                            | 16,6      |
| 12,5  | -28,480              | 0,049                                                                                                                         | -130,655          | 0,267             | 0,790  | 0,301 | 0,483                            | 17,0      |
| 17,5  | -27,630              | 0,045                                                                                                                         | -133,604          | 0,653             | 0,816  | 0,701 | 0,538                            | 20,2      |
| 32,5  | -27,969              | 0,168                                                                                                                         | -136,375          | 1,441             | 0,814  | 0,813 | 0,593                            | 23,1      |
| 52,5  | -27,802              | 0,009                                                                                                                         | -135,387          | 1,719             | 0,820  | 0,526 | 0,538                            | 20,2      |
| 67,5  | -28,063              | 0,003                                                                                                                         | -133,719          | 0,800             | 0,790  | 0,732 | 0,623                            | 24,5      |
| 77,5  | -28,286              | 0,026                                                                                                                         | -129,138          | 2,394             | 0,777  | 0,098 | 0,477                            | 16,6      |
| 107,5 | -27,724              | 0,006                                                                                                                         | -136,198          | 0,822             | 0,789  | 0,132 | 0,484                            | 17,1      |
| 112,5 | -27,446              | 0,118                                                                                                                         | -133,986          | 0,155             | 0,801  | 0,306 | 0,481                            | 16,9      |
| 115   | -27,449              | 0,004                                                                                                                         | -126,742          | 0,999             | 0,824  | 0,193 | 0,488                            | 17,3      |
| 122,5 | -27,781              | 0,295                                                                                                                         | -131,669          | 0,458             | 0,806  | 0,094 | 0,409                            | 12,0      |
| 130   | -27,714              | 0,040                                                                                                                         | -127,357          | 0,338             | 0,793  | 0,274 | 0,447                            | 14,7      |
| 142,5 | -27,757              | 0,175                                                                                                                         | -131,973          | 0,225             | 0,800  | 0,194 | 0,492                            | 17,5      |
| 152,5 | -27,696              | 0,089                                                                                                                         | -134,566          | 1,495             | 0,775  | 0,101 | 0,482                            | 16,9      |
| 155   | -27,461              | 0,054                                                                                                                         | -133,870          | 0,728             | 0,782  | 0,224 | 0,444                            | 14,5      |
| 162,5 | -27,871              | 0,046                                                                                                                         | -136,287          | 0,535             | 0,768  | 0,343 | 0,524                            | 19,4      |
| 165   | -26,706              | No value                                                                                                                      | -130,842          | 1,740             | 0,775  | 0,229 | 0,448                            | 14,8      |
| 172,5 | -27,323              | 0,161                                                                                                                         | -133,485          | 0,921             | 0,770  | 0,156 | 0,462                            | 15,7      |
| 180   | -27,454              | 0,001                                                                                                                         | -135,430          | 0,911             | 0,775  | 0,313 | 0,456                            | 15,2      |
| 187,5 | -27,633              | 0,149                                                                                                                         | -136,050          | 0,655             | 0,762  | 0,309 | 0,496                            | 17,8      |
| 207,5 | -27,430              | 0,222                                                                                                                         | -136,687          | No value          | 0,785  | 0,063 | 0,477                            | 16,6      |
| 287,5 | -27,459              | No value                                                                                                                      | -137,911          | 2,688             | 0,803  | 0,126 | 0,460                            | 15,6      |
| 327,5 | -27,475              | 0,072                                                                                                                         | -139,901          | 1,058             | 0,769  | 0,117 | 0,474                            | 16,4      |
| 397,5 | -27,213              | 0,027                                                                                                                         | -141,268          | 0,301             | 0,765  | 0,150 | 0,463                            | 15,7      |
| 452,5 | -27,610              | 0,023                                                                                                                         | -138,820          | 1,432             | 0,781  | 0,159 | 0,471                            | 16,2      |
| 482,5 | -27,626              | No value                                                                                                                      | -142,785          | No value          | 0,754  | 0,100 | 0,471                            | 16,3      |
| 487,5 | -27,181              | 0,023                                                                                                                         | -137,367          | 0,767             | 0,749  | 0,087 | 0,471                            | 16,3      |

Table 3 GeoB18308-1 organic geochemical downcore data. *n*-Alkane isotopic composition and distribution descriptive parameters averaged (with standard deviation), BIT,  $\text{TEX}_{86}^{\text{H}}$  and  $\text{SSTs}(^{\circ}\text{C})$  calculated from  $\text{TEX}_{86}^{\text{H}}$  after Kim et al. (2010).

**Table 4 Mineralogical composition of heavy minerals in silt and sand fraction (15-500 $\Box$ m) of marine sediments from GeoB18308-1 and Gouritz River catchment**

| location       | sample name | HM %weight | zircon | tourmaline | rutile | titanite | apatite | epidote | garnet | chloritoid | sillimanite | amphibole | clinopyroxene | orthopyroxene | Total | ZTR | <pre>% transparent</pre> | % opaque | % Fe oxides | % Ti oxides | % turbid HM | Total |
|----------------|-------------|------------|--------|------------|--------|----------|---------|---------|--------|------------|-------------|-----------|---------------|---------------|-------|-----|--------------------------|----------|-------------|-------------|-------------|-------|
| Gouritz River: |             |            |        |            |        |          |         |         |        |            |             |           |               |               |       |     |                          |          |             |             |             |       |
| Bland´s Drift  | ECT2-1      | 1,5        | 4      | 2          | 1      | 5        | 6       | 9       | 7      | 0          | 0           | 4         | 62            | 1             | 100   | 6   | 67%                      | 8%       | 15%         | 9%          | 1%          | 100%  |
| Mossel Bay 5km | GeoB18308-1 |            |        |            |        |          |         |         |        |            |             |           |               |               |       |     |                          |          |             |             |             |       |
| offshore       | (11 cm)     | 1,3        | 6      | 4          | 5      | 3        | 6       | 19      | 5      | 1          | 0           | 5         | 46            | 2             | 100   | 15  | 72%                      | 10%      | 7%          | 11%         | 0%          | 100%  |
| Mossel Bay 5km | GeoB18308-1 |            |        |            |        |          |         |         |        |            |             |           |               |               |       |     |                          |          |             |             |             |       |
| offshore       | (285 cm)    | 1,3        | 7      | 2          | 2      | 3        | 4       | 10      | 7      | 0          | 0           | 2         | 60            | 2             | 100   | 12  | 77%                      | 10%      | 1%          | 12%         | 0%          | 100%  |

Fig. 1 Schematic overview of the main components of the oceanic (thick arrows) and atmospheric (thin arrows) circulation over southern Africa (modified from Truc et al., 2013). The Winter Rainfall Zone is colored in red; the Summer Rainfall Zone in green and the year-round rainfall zone in yellow. Squares indicate localities of Gouritz River samples; diamond indicates marine sediment core location. SHW = Southern Hemispheric Westerlies; ITCZ = Intertropical Convergence Zone. Note that these are shown in their summer position. Austerlitz midden (1), Katbakkies Pass midden (2), Seweweekspoort midden(3), Pinnacle Point (4), Cango Cave (5), Nelson Bay Cave (6), Braamhoek peat (7), Lake Eteza (8), Cold Air Cave (9), Wonderkrater (10)

Fig. 2 The map is an inlet of Fig. 1 and shows vegetation types in and around the drainage basin (after Mucina and Rutherford, 2006 and Scott et al., 2012) of the Gouritz River. Squares indicate localities of Gouritz River samples; diamond indicates marine sediment core location. Plots showing regional biome/eco-region soil and vegetation average n-alkane distributions from Carr et al. (2015) are indicated on the left whereas average n-alkane distributions of riverine and marine samples from this study are plotted above the map.

Fig. 3 on left: Core log of GeoB18308-1 (internal structures with recorded colors) demonstrating the presence of an event deposit (blue box). On right: the calibrated 14C dates (transparent blue) and the age-depth model (darker greys indicate more likely calendar ages; grey stippled lines show 95% confidence intervals; red curve shows single 'best' model based on the weighted mean age for each depth). The green box marks horizons of increased terrestrial input where a terrestrial calibration curve (SHCal13: Southern Hemisphere (Hogg et al., 2013)) was used. Otherwise, Marine13-modelled ocean average (Reimer et al., 2013) with a  $\delta R$  following Dewar et al. (2012) and Meadows et al. (in prep/personal communication). Redeposited material (i.e. a crab claw at 123 cm depth, a bivalve at 285 cm and the reworked package at 30 – 60 cm depth) were excluded from the age-depth modeling. Medieval Climate Anomaly (MCA) and the Little Ice Age (LIA) are indicated.

Fig. 4 Organic and inorganic down-core geochemistry of GeoB18308-1 (a-g) as indicators for fluvial input, SST, SRZ and aridity. Low resolution records are marked by colored dots. SSTs for samples with BIT above 0.3 were not calculated. The indicator of estuarian inflow is deduced from factor loadings along axes 1 of a PCA on microfossil distribution of core samples. For comparative purposes regional paleoenvironmental records are plotted: h) aridity index from Seweweekspoort (Chase et al., 2013) and i) indirect humidity index from Groenvlei (Wündsch et al., 2016).

Fig. 5 Variations in  $\delta D$  of the n-C31-alkane [%VSMOW] in the distinct horizons of paleoflood vs. soil formation horizons. Gouritz River tributary cut bank at sample location ECT-3 is depicted as an example illustrating these alternating horizons.

Fig. 6 Conceptual model after Cohen and Tyson (1995) depicting the recorded response (in red) to shifts in atmospheric (indicated in black) and oceanic (in blue) circulation patterns during the LIA (a) and MCA (b). SIA (=South Indian Anticyclone), SAA (=South Atlantic Anticyclone), SHW (=Southern Hemispheric Westerlies), AC (=Agulhas Current), LIA (=Little Ice Age), MCA (=Medieval Climate Anomaly), ITCZ = Intertropical Convergence Zone.